# Construction of Metastasis Prediction Models and Screening of Anti-Metastatic Drugs Based on Pan-Cancer Single-Cell EMT Features

**DOI:** 10.3390/ijms262311582

**Published:** 2025-11-29

**Authors:** Yingqi Xu, Yawen Luo, Maohao Li, Na Lv, Yuanyuan Deng, Ning Li, Shichao Wan, Xing Gao, Xia Li, Congxue Hu

**Affiliations:** College of Bioinformatics Science and Technology, Harbin Medical University, Harbin 150081, China; lyw009031@163.com (Y.L.); hmu_limaohao@163.com (M.L.); 15944936801@163.com (N.L.); dyy2439710811@163.com (Y.D.); 13668825975@163.com (N.L.); 19989106390@163.com (S.W.); gx011018@hrbmu.edu.cn (X.G.); lixia@hrbmu.edu.cn (X.L.)

**Keywords:** epithelial–mesenchymal transition, single-cell RNA sequencing, metastasis prediction model, drug screening

## Abstract

Tumor metastasis is the leading cause of death in cancer patients, with epithelial–mesenchymal transition (EMT) playing a key role. To systematically elucidate the cellular mechanisms and molecular networks through which EMT drives metastasis across cancers, this study integrated transcriptomic data from over 1.2 million single cells across 265 samples representing 12 primary epithelial cancers, constructing a comprehensive pan-cancer single-cell atlas covering diverse stages and metastatic states. By analyzing the metastatic features and interaction networks of malignant epithelial cells and cancer-associated fibroblasts (CAFs), we identified cancer-specific metastasis-related gene sets. Based on these genes, multiple machine learning algorithms were applied to build cancer-specific and cross-cancer metastasis prediction models, leading to the development of the metastasis prediction score (MPS) and global metastasis prediction score (GMPS). Both scores showed excellent predictive performance in independent test and external validation cohorts. MPS exhibited higher cancer specificity, whereas GMPS showed stronger cross-cancer generalization. Moreover, elevated MPS and GMPS reflected immunosuppressive tumor microenvironment features and were significantly associated with poor prognosis across multiple cancer types. Finally, through a drug repositioning framework, we identified several potential anti-metastatic compounds targeting the metastasis network, among which Fostamatinib demonstrated broad-spectrum therapeutic potential against metastasis in multiple cancers.

## 1. Introduction

Tumor metastasis is the leading cause of cancer-related mortality, with over 90% of cancer deaths resulting from the dissemination of tumors rather than the primary lesion itself [1]. This process involves a multi-step cascade, including detachment of cancer cells from the primary site, invasion into surrounding tissues, the breaching of the basement membrane, entry into the blood or lymphatic circulation, and colonization and growth at distant organs [2,3]. Despite significant advances in cancer diagnosis and treatment in recent years, managing metastatic disease remains a formidable challenge, and patient prognosis is generally poor [4,5]. Studies indicate that the 5-year survival rate of patients initially diagnosed with metastatic cancer has shown minimal improvement [6]. The core difficulty in addressing tumor metastasis lies in the incomplete understanding of its molecular mechanisms, which involve processes such as epithelial–mesenchymal transition, acquisition of stem cell-like properties, and reprogramming of key signaling pathways [7,8,9]. Therefore, an in-depth investigation of the molecular mechanisms underlying tumor metastasis is crucial not only for elucidating the malignant progression of cancer and identifying potential metastasis-related molecules, but also for providing a theoretical basis for early intervention in high-risk patients, the development of novel anti-metastatic therapeutics, and the optimization of treatment strategies, ultimately aiming to improve patient survival and prognosis.

Epithelial–mesenchymal transition (EMT) is recognized as a central biological process driving tumor metastasis [10,11,12], typically occurring in the early stages of metastasis in epithelial cancers [12]. EMT enhances cancer cell motility, invasiveness, and therapeutic resistance by promoting the loss of epithelial characteristics and the acquisition of mesenchymal traits [13]. This process involves the activation of multiple transcription factors, including members of the SNAIL, ZEB, and TWIST families, as well as the coordinated regulation of downstream signaling networks [14,15]. EMT is closely associated with stem cell-like properties [16], immune evasion [17], and remodeling of the tumor microenvironment [18]. During EMT, cell–cell adhesion is reduced, apical-basal polarity is disrupted, and migratory capacity is acquired, thereby facilitating the progression from benign lesions to invasive cancers. Notably, EMT frequently precedes overt metastasis, and EMT-related features in patients can serve as important predictors of metastatic potential, providing critical guidance for early intervention and prognostic assessment [19,20]. Previous studies have developed predictive models of metastasis and prognosis in cancers such as breast cancer, lung cancer, and head and neck squamous cell carcinoma based on EMT characteristics [21,22,23], demonstrating their utility as effective risk assessment tools. However, these studies are mostly limited to analyses of a single cancer type or of metastatic lesions, and pan-cancer models for predicting metastasis from the perspective of primary tumors remain relatively scarce.

Malignant epithelial cells serve as the primary drivers of EMT, and their state transitions directly dictate tumor metastatic potential [24,25]. Within primary tumors, these cells are highly heterogeneous [26], with certain subsets exhibiting pronounced EMT characteristics and enhanced invasive and migratory capabilities [27]. Such cells typically display downregulation of the epithelial marker E-cadherin (CDH1), upregulation of mesenchymal markers including N-cadherin (CDH2) and vimentin (VIM), and phenotypic changes such as cytoskeletal reorganization, altered cell morphology, and loss of polarity [28,29]. Recent studies have demonstrated that EMT is not a binary ‘all-or-nothing’ process; rather, multiple intermediate states exist. Cells in these intermediate states co-express epithelial and mesenchymal markers, exhibiting heightened plasticity and adaptability, which may further facilitate metastasis and colonization [30]. Therefore, analyzing the dynamic evolution and functional heterogeneity of malignant epithelial cells during EMT is crucial for identifying key driving events and potential therapeutic targets.

Cancer-associated fibroblasts (CAFs) represent one of the most important stromal cell populations within the tumor microenvironment. In recent years, CAFs have been recognized as key players in EMT and tumor metastasis [31,32]. Accumulating evidence indicates that CAFs not only express EMT-related genes themselves but also exhibit pronounced EMT characteristics and actively regulate these genes within the tumor microenvironment [32,33]. During EMT, many marker genes, such as VIM and FSP1, are co-expressed by CAFs alongside epithelial cells [34]. Moreover, EMT-associated transcription factors are highly expressed in CAFs [35]. The transformation of epithelial cells into CAFs via EMT has been extensively validated across multiple studies [36,37,38,39]. For instance, in renal fibrosis, myofibroblasts (myCAFs) expressing epithelial markers were observed under specific culture conditions [40]. In breast cancer, intermittent hypoxia can induce CAFs to undergo mesenchymal–epithelial transition (MET), gradually acquiring a more invasive epithelioid phenotype [36]. Furthermore, epithelial-to-CAF transdifferentiation via EMT has been confirmed in esophageal and colorectal cancers [38,39]. Collectively, epithelial cells and CAFs not only play central roles in establishing EMT characteristics but also critically contribute to tumor metastasis, potentially exhibiting similar differentiation behaviors across other epithelial cancers. Therefore, an in-depth analysis of the reciprocal transformation between epithelial cells and CAFs, and its role in the EMT process, will provide a crucial scientific foundation and translational potential for developing novel strategies targeting the tumor microenvironment to inhibit cancer metastasis.

Single-cell RNA sequencing (scRNA-seq) enables the analysis of cellular heterogeneity in solid tumors and reveals how epithelial cells and fibroblasts in the tumor microenvironment influence tumor initiation, progression, and metastasis through cellular and molecular mechanisms [41]. In this study, we integrated pan-cancer single-cell transcriptomic data to construct a primary pan-cancer single-cell atlas encompassing different tumor stages and metastatic states. This analysis systematically delineated metastasis-associated features and cell–cell interaction networks of malignant epithelial cells and CAF subtypes, and identified cancer-specific metastasis-signature gene sets. Based on these gene signatures, we developed a predictive model for tumor metastasis and screened potential anti-metastatic compounds. Overall, our study provides a comprehensive understanding of tumor metastasis from the cellular to molecular level, offering a theoretical foundation and valuable resources for precise therapeutic intervention.

## 2. Results

### 2.1. Pan-Cancer Single-Cell Atlas Reveals Metastasis Signatures in the Tumor Microenvironment

To establish a comprehensive single-cell atlas covering various human tissues across different cancer stages and metastatic states, we collected and integrated 32 publicly available single-cell transcriptome datasets, encompassing 12 common primary human epithelial cancers, with a total of 265 tumor samples (Appendix A). These cancers include breast carcinoma (BC), bladder carcinoma (BLCA), clear cell renal cell carcinoma (ccRCC), colorectal carcinoma (CRC), esophageal squamous cell carcinoma (ESCC), gastric carcinoma (GC), hepatocellular carcinoma (HCC), head and neck squamous cell carcinoma (HNSCC), lung carcinoma (LC), ovarian carcinoma (OC), pancreatic ductal adenocarcinoma (PDAC), and prostate adenocarcinoma (PRAD). Based on clinical information, samples were stratified into four TNM stages (I-IV) and further classified by metastatic status into non-metastasis (NM), regional metastasis (RM, primarily lymph node or peritoneal metastasis), distant metastasis (DM), and unknown metastasis (UM). Using these scRNA-seq data, we systematically identified genes closely associated with epithelial–mesenchymal transition (EMT) across different cancers and, in combination with TCGA RNA-seq data, constructed cancer-specific metastasis risk prediction models. By integrating drug-target interaction networks with CMap (Connectivity Map) drug-induced gene expression data, candidate drugs potentially associated with metastasis were identified. The overall analytical workflow of this study is illustrated in Figure 1A.

After rigorous quality control, data integration, and gene marker annotation, a pan-cancer single-cell transcriptome map comprising 1,217,505 cells was generated (Figure 1B,C and Appendix A). Using established cell type marker genes, we identified eight major cell populations: epithelial cells (N = 295,384), B cells (N = 73,171), plasma cells (N = 88,976), T/NK cells (N = 393,963), myeloid cells (N = 186,726), mast cells (N = 15,308), fibroblasts (N = 101,093), and endothelial cells (N = 62,884) (Appendix A). The top 10 characteristic genes of each cell type were then extracted for Gene Ontology (GO) annotation, revealing enriched biological functions that closely matched the known characteristics of each cell type, confirming the accuracy of cell annotations (Appendix A).

In order to further identify malignant cells in the epithelial cell population, the inferCNV tool was used to calculate copy number variation (CNV) values for each cell, with T/NK cells as a reference. Epithelial cells exhibited extensive copy number alterations, including amplifications and deletions, indicative of their pronounced malignant features across different cancer types (Appendix A). To further identify malignant cells within the epithelial population, we calculated CNV Deviation based on CNV Scores to quantify the overall deviation of each cell’s copy number profile from the neutral state. Accordingly, epithelial cells exhibited elevated CNV Deviation (Appendix A). We computed the Pearson correlation coefficient between each cell and the mean CNV Deviation of the top 5% of cells. Thresholds for classification were determined based on the distributions of CNV Deviation and correlation values. The histograms show that the number of cells to the right of the selected thresholds is substantially higher than that on the left, resulting in a clear distributional separation (Appendix A). Cells exhibiting CNV Deviation > 0.002 and correlation > 0.1 were classified as malignant. Applying this approach, we successfully identified a subset of malignant cells within the epithelial population (Appendix A). Next, we examined the distribution of malignant and non-malignant epithelial cells across cancer types. We found that the proportion of malignant epithelial cells was relatively low in BLCA and ESCC. Upon further inspection of the sample sources, we determined that this pattern was primarily due to the absence of late-stage (particularly stage IV) or distantly metastatic samples in these datasets. Thus, the lower abundance of malignant cells is consistent with the sample composition and, to some extent, supports the reliability of our malignancy classification (Appendix A).

Malignant cells were incorporated into the full cell map for integrated analysis. We found that the distribution of cell types exhibited pronounced heterogeneity across different samples of each cancer type (Figure 1B and Appendix A; Appendix A). The proportion of malignant cells gradually increased with advancing patient stage (Appendix A) and was significantly enriched in distant metastasis samples (Figure 1D and Appendix A), indicating that Malignant cell expansion is closely associated with tumor progression and metastasis.

To explore the sources of EMT characteristics beyond epithelial cells, the EMT gene set was used to score various cell types. We observed that EMT scores increased significantly with tumor progression in the majority of cancer types (Appendix A). Remarkably, Fibroblasts exhibited the highest EMT scores among all populations (Figure 1E and Appendix A), indicating that both epithelial cells and fibroblasts in the tumor microenvironment display EMT features.

Intercellular communication analysis using the CellChat tool revealed that interactions between fibroblasts and malignant cells were the most frequent (Figure 1F and Appendix A). Notably, these interactions remained the most prominent across different metastatic states and progressively increased with advancing metastasis (Figure 1F,G). Pathway analysis further showed that these interactions primarily involved key signaling pathways related to extracellular matrix remodeling and metastasis, such as LAMININ and COLLAGEN, with both the number and intensity of interactions rising significantly as metastasis progressed (Appendix A). These findings underscore the central role of fibroblasts in driving tumor progression and metastasis.

### 2.2. Phenotypic Characterization and Metastasis-Associated Feature Identification of Malignant Epithelial Cell Subgroups

Since EMT primarily occurs in malignant epithelial cells, these cells were first extracted for in-depth analysis. Re-clustering and UMAP (uniform manifold approximation and projection) visualization according to cancer type revealed pronounced heterogeneity among malignant cells from different sources (Appendix A), warranting independent analyses for each cancer. Comparison of EMT scores across stages and metastatic states revealed that EMT activity was relatively low in early stages (I–II), substantially higher in late stages (III–IV), and consistently elevated in metastatic samples compared with non-metastatic samples, highlighting a robust association between EMT and metastasis (Figure 2A and Appendix A). Based on EMT scores, malignant cells were classified into three subtypes, high EMT (hEMT), medium EMT (mEMT), and low EMT (lEMT), using the 60% and 40% quantiles as thresholds (Figure 2B and Appendix A). Analysis of the proportion of each subtype across stages and metastatic states revealed a gradual increase in hEMT cells in late-stage samples (III–IV) (Figure 2C and Appendix A). To further validate the biological relevance of this classification, cancer stem cell (CSC) scores were calculated using common CSC markers (EPCAM, CD24, KRT19, SOX9, PROM1, CD44, THY1, CD47 [42,43,44,45]) (Appendix A). We then assessed the correlation between CSC and EMT scores, which revealed a strong and significant positive association (Appendix A). Moreover, hEMT cells exhibited significantly higher CSC scores compared to other EMT subtypes (Appendix A), consistent with their potential metastatic capacity.

Subsequently, the biological characteristics of the three EMT subtypes were systematically compared. Analysis of cell cycle status revealed that in most cancers, hEMT cells were predominantly in G1 phase, whereas lEMT cells were mainly in G2/M phase (Appendix A). These findings suggest that cells with high EMT activity are in a ‘migration-invasion’ state, temporarily pausing proliferation while preparing for migration and executing epithelial–mesenchymal transition, whereas low EMT cells adopt a ‘proliferative’ state with active division. This observation aligns with previous studies, supporting a mutually restrictive relationship between EMT and the cell cycle [46,47].

To investigate the functional differences among EMT subtypes, GSVA analysis was performed. hEMT cells exhibited significantly higher activity in pathways associated with cancer progression, including the TGF-β signaling pathway and cell migration, and were also markedly enriched in inflammation- and stress-related pathways, such as IL6/JAK/STAT3 signaling, hypoxia, angiogenesis, and autophagy-related pathways, all closely linked to metastasis. Notably, hEMT cells also showed relatively high scores in epigenetic regulation pathways, particularly those involving histone deacetylases (HDACs) and histone acetyltransferases (HATs). These findings suggest that epigenetic remodeling may cooperate with EMT programs to enhance chromatin accessibility, stabilize mesenchymal transcriptional states, and promote the acquisition of migratory and invasive phenotypes, thereby contributing to metastatic potential [48,49]. In contrast, lEMT cells were predominantly enriched in proliferation-related pathways (Figure 2D), consistent with previous cell cycle analysis.

Metabolic pathway analysis further revealed distinct profiles between EMT subtypes. hEMT cells were significantly enriched in glucose metabolism and specific lipid metabolic pathways, including glycolysis and sphingolipid metabolism, reflecting metabolic reprogramming: enhanced glycolysis provides rapid energy and metabolic intermediates to support migration and invasion in highly metastatic cells, while increased sphingolipid metabolism facilitates membrane remodeling and signal transduction, promoting cell motility and adaptation to the tumor microenvironment [50,51]. Conversely, lEMT cells exhibited higher activity in fatty acid metabolism, nucleotide metabolism, energy metabolism, and drug metabolism pathways, indicating their tendency to maintain homeostatic metabolism and support proliferation-related metabolic processes (Figure 2D and Appendix A).

To investigate cell state evolution, we combined CytoTRACE and Monocle to infer the development trajectory. Specifically, CytoTRACE was used to score the differentiation status of single cells, and cells with low differentiation scores were designated as the starting point for Monocle trajectory inference, reconstructing the pseudo-temporal progression of malignant cells. Trajectory analysis revealed that, over pseudo-time, cells gradually transitioned from lEMT to hEMT (Figure 2E and Appendix A), suggesting that tumor cells may dynamically evolve from low to high metastatic potential during cancer progression. Comparative analysis of differentiation potential among EMT subtypes showed that hEMT cells generally exhibited stronger differentiation capacity, whereas lEMT cells were comparatively weaker (Figure 2F). Notably, cells with high differentiation potential often display greater plasticity and stemness [52], which may confer enhanced adaptability and a pivotal role in driving tumor progression and metastasis.

To comprehensively characterize dysplastic features in malignant cells, we applied non-negative matrix factorization (NMF) to systematically extract transcriptional heterogeneity within each cancer type. Specifically, highly variable genes from each sample were decomposed, and the optimal rank was determined to obtain stable gene modules. Across cancer types, 70–322 modules were identified per cancer, yielding a total of 2004 modules. Through cross-sample integration, gene co-occurrence network filtering, and robust module detection using the Infomap algorithm, 6–7 representative modules were identified per cancer type. After merging highly similar modules, a final set of 13 modules was obtained (Figure 2G). Functional annotation of these modules, performed through HALLMARK and GO enrichment analyses (Appendix A), revealed that each module represented a distinct ‘functional program’. Among these, five core programs were consistently identified across all cancers: Program 1 (ATF3, DDIT4, etc.), associated with stress; Program 2 (CD64, IFITM3, etc.), related to immune regulation; Program 3 (RPL10, RPL13A, etc.), linked to ribosome; Program 4 (COL1A1, LAMA3, etc.), indicative of metastasis; and Program 5 (CCNB1, MKI67, etc.), associated with proliferation. In addition, five programs were present only in subsets of cancer types, including metabolism (Program 6: ACER2, ALDOB, etc.), epithelial differentiation (Program 7: KRT5, S100A7, etc.), oxidative phosphorylation (Program 8: ATP5F1B, COX4I1, etc.), protein synthesis (Program 9: HSPA5, HSPA8, etc.), and RNA processing (Program 10: CCNL1, DDX17, etc.). Importantly, three programs demonstrated cancer-type specificity: pulmonary function program in lung cancer (Program 11: ANXA2, HOPX, etc.), inflammatory response program in colorectal cancer (Program 12: ALOX5, ALOX5AP, etc.), and cell adhesion program in hepatocellular carcinoma (Program 13: ADAM17, CDC42, etc.) (Figure 2G and Appendix A).

Then, we applied the functional program gene sets identified by NMF to score the three EMT subtypes and visualized their distributions using UMAP. hEMT cells exhibited markedly higher scores in stress response, immune regulation, and metastasis programs, indicating stronger adaptability and invasive capacity. These features may enable hEMT cells to more effectively respond to tumor microenvironmental stress signals and to facilitate immune evasion and metastatic dissemination, consistent with their clinical association with high metastatic potential and poor prognosis [53,54,55]. By contrast, lEMT cells showed higher scores in ribosome- and proliferation-related programs, in line with our previous findings (Figure 2H and Appendix A).

To further delineate the molecular characteristics associated with tumor metastasis, we performed weighted gene co-expression network analysis (WGCNA) on hEMT cells across different cancer types and identified a series of co-expression modules within each cancer. We then calculated the correlations between each WGCNA module and the functional programs identified by NMF (Figure 2I and Appendix A) (Appendix A). Based on the threshold (|r| > 0.2, *p* < 0.05), modules significantly associated with metastasis-related programs were screened, and their corresponding gene sets were extracted to generate the first set of metastasis-associated gene signatures (GeneSet1) (Appendix A).

In parallel, we employed a mapping-based differential gene analysis strategy to compare hEMT and lEMT cells across cancer types. Specifically, both groups were independently clustered, followed by anchor-based mapping to establish molecular correspondence between the two cell states. After mapping, highly similar subsets were systematically removed to minimize confounding signals, and differential expression analysis was subsequently performed on the remaining cells (Figure 2J and Appendix A). Using a significance threshold (*p* < 0.05, |avg-log2FC| > 0.25), we derived a second set of metastasis-associated gene signatures (GeneSet2) (Appendix A). This multi-tiered strategy markedly enhanced the sensitivity of detecting differentially expressed genes between high- and low-metastatic potential cells and provided a more refined molecular signature landscape for elucidating the mechanisms of tumor metastasis.

### 2.3. Characterization of Fibroblast Subtypes and Dynamic Intercellular Communication with Malignant Epithelial Cells

Based on the analysis of previous results, we observed that fibroblasts exhibited the highest EMT scores within the tumor microenvironment, representing a major source of EMT-related gene expression in epithelial cancers and showing significant interactions with malignant epithelial cells. To further elucidate the role of fibroblasts in tumor progression, we conducted a detailed analysis.

First, a total of 101,093 fibroblasts were re-clustered and classified according to canonical marker genes. The representative markers used were as follows: IGF1 and C7 for normal fibroblasts (NFs), MYH11 and RGS5 for smooth muscle cells/pericytes, CXCL8 and APOE for inflammatory cancer-associated fibroblasts (iCAFs), HLA-DPA1 and CD74 for antigen-presenting cancer-associated fibroblasts (apCAFs), and COL3A1 and FN1 for myofibroblastic cancer-associated fibroblasts (myCAFs). This classification delineated five distinct fibroblast subpopulations (Figure 3A and Appendix A). Subsequently, the top 10 highly expressed genes from each subtype were extracted and subjected to GO annotation, which further confirmed the accuracy of the cell-type classification (Appendix A).

Next, we examined the proportional distribution of cell subtypes across different stages and metastatic states. After excluding smooth muscle cells and pericytes, we found that iCAFs and myCAFs were the predominant CAF subtypes in tumor tissues, with their proportions increasing alongside cancer progression (Figure 3B and Appendix A). Functionally, myCAFs promote tumor invasion by enhancing ECM deposition and remodeling matrix mechanics, whereas iCAFs modulate the immune microenvironment by secreting inflammatory factors, such as IL6 and CXCL12, to support tumor growth [56,57]. Further analysis of EMT characteristics revealed that myCAFs exhibited significantly higher EMT scores across all cancers, highlighting their key role in cancer progression and metastasis (Figure 3C).

Subsequently, we applied GSVA to assess the pathway enrichment of CAF subtypes in metastasis-associated signaling. The results revealed that myCAFs were strongly enriched in pathways related to metastasis, including WNT, hypoxia, NOTCH, and collagen fibril organization. These pathways regulate tumor cell proliferation, invasion, and matrix interactions, suggesting that myCAFs may facilitate metastasis by remodeling the tumor microenvironment. In contrast, other CAF subtypes exhibited relatively low activity in these pathways (Figure 3D). Integrating these findings, we observed that both hEMT cells and myCAFs displayed high EMT scores and shared similar enrichment profiles across multiple metastasis-related pathways, indicating a potential developmental relationship between the two populations. To further investigate these interactions, we performed a joint analysis of three EMT subtypes and three CAF subtypes. Cell–cell communication analysis demonstrated that myCAFs exerted the most pronounced influence on hEMT cells (Figure 3E and Appendix A). Pathway-specific analyses revealed that myCAFs strongly interacted with hEMT cells through canonical signaling axes such as THBS, FN1, and COLLAGEN (Figure 3F and Appendix A). Moreover, in the TGFβ pathway, hEMT cells exhibited significant regulatory effects on myCAFs (Appendix A). This reciprocal signaling is known to drive EMT, remodel the tumor microenvironment of malignant epithelial cells, and enhance metastatic potential [58].

Furthermore, we applied pySCENIC to systematically analyze the specific transcription factors of three EMT subtypes and three CAF subtypes, selecting the top 30 factors with the highest RSS values in each cancer type. After integrating cross-cancer distributions, transcription factors recurrent in ≥4 cancer types were retained as potential shared regulatory factors. The results showed that myCAFs were enriched for TWIST1, ZEB1, and LEF1. TWIST1 and ZEB1 are core regulators of EMT, capable of repressing E-cadherin and promoting cell migration and invasion [59], while LEF1 is a key transcription factor in the WNT/β-catenin pathway, involved in matrix remodeling and metastasis [60]. In contrast, hEMT cells predominantly expressed FOSB, SOX9, and ATF3, which are closely associated with inflammatory responses and the maintenance of stemness [61,62,63]. Notably, myCAFs and hEMT cells exhibited cross-expression of certain characteristic transcription factors (e.g., EGR2, ETS1), suggesting a high degree of convergence between the two populations in regulating metastasis-related processes (Figure 3G).

On this basis, we integrated and re-clustered hEMT cells and myCAFs across different cancers (Appendix A) and performed trajectory inference using Slingshot. The results indicated a gradual transition from hEMT cells toward myCAF, suggesting a potential continuous evolutionary relationship between the two populations. Along the pseudotime trajectory, hEMT cells progressively lost epithelial characteristics (e.g., decreased CDH1 expression), accompanied by increased expression of mesenchymal-associated genes and ECM remodeling signals (e.g., elevated VIM expression), ultimately converging toward a myCAF state (Figure 3H and Appendix A).

Spatial transcriptomics (ST) data further supported the above findings. We analyzed ST data from patients with BC and confirmed the spatial organizational structure. Using Leiden clustering and spot feature analysis, we divided the tissue into 15 subgroups. We then applied the CRAD tool to project scRNA-seq data from all previously annotated cell types onto these ST-captured spots, where transcriptionally similar spots displayed corresponding signals on the spatial map. Subgroup 2 was identified as hEMT cells and subgroup 13 as myCAFs. We next extracted these two subgroups for trajectory analysis and used stlearn to infer their spatial transcriptional dynamics. The results showed a directional flow from subgroup 2 (hEMT) toward subgroup 13 (myCAF) (Figure 3I and Appendix A), verifying the EMT of hEMT epithelial cells into myCAFs. To assess the generalizability of this phenomenon, we further analyzed ST datasets from ccRCC and PDAC. Consistently, we observed spatial trajectories of hEMT cells transitioning into myCAFs (Appendix A). Notably, in ccRCC, we clearly detected the co-localization of hEMT cells and myCAFs in spatial projections of scRNA-seq data onto ST, once again confirming their close spatial association and potential transition relationship within the tumor microenvironment (Appendix A).

To systematically analyze the dynamic gene regulation underlying the transformation process, we leveraged the Slingshot trajectory to screen significant genes expressed in sufficient cells as ordination genes. Using the pseudotime inferred by Slingshot, we then performed dynamic transcriptional program analysis. A series of characteristic genes (*p* < 0.05, |τ| > 0.4) closely associated with pseudotime were subsequently identified through the Mann–Kendall trend test. The pseudotime heatmap revealed two major gene clusters: genes upregulated in the early stage of EMT were primarily involved in protein synthesis and cell proliferation, providing essential conditions for subsequent tumor metastasis, whereas genes upregulated in the late stage were significantly enriched in ECM remodeling, cell adhesion, and other pathways linked to mesenchymal cell functions, reflecting the canonical dynamic process of EMT (Appendix A). Based on this analysis, we defined the third set of metastasis-associated gene signatures, referred to as GeneSet3 (Appendix A).

Based on the three gene sets obtained in Results 2 and 3, we intersected them across different cancers and identified two groups of genes—those upregulated and those downregulated during EMT in each cancer (Figure 3J; Appendix A). We then examined the overlap of these genes among cancers. The string diagram highlighted genes consistently upregulated or downregulated in three or more cancer types (Appendix A). To further explore the biological significance of these differentially expressed genes, we extracted those shared by at least two cancers and conducted GO enrichment analysis. The results revealed that upregulated genes were significantly enriched in pathways closely associated with tumor metastasis, including cell adhesion and migration (e.g., cell–cell junction assembly, epithelial cell migration), extracellular matrix organization, Wnt signaling, and stem cell population maintenance. In contrast, downregulated genes were predominantly enriched in pathways related to metabolism and fundamental cellular processes, such as energy metabolism and biosynthesis (e.g., ATP synthesis coupled electron transport), gene expression and protein processing (e.g., rRNA processing), as well as stress responses and apoptosis-related functions (e.g., cellular response to toxic substances) (Figure 3K).

### 2.4. Construction of Metastasis Prediction Models and Analysis of Their Immunological Features and Prognostic Implications

In our previous single-cell analysis, we identified sets of cancer-specific metastasis-signature genes that elucidated key molecular mechanisms underlying metastasis across different cancer types. To systematically assess the predictive value of these genes and to screen for key discriminative markers, the analysis was extended to the bulk RNA-seq level. We integrated RNA-seq data from 12 cancer types obtained from the TCGA and GEO databases. After stringent quality control, a total of 2992 samples with complete clinical information were retained, comprising 1407 non-metastasis and 1585 metastasis cases (Appendix A).

Considering the heterogeneity among different cancer types in terms of tissue origin, driver mutations, and tumor microenvironment, we constructed cancer-specific metastasis models to improve predictive accuracy and clinical applicability. The workflow is as follows: first, the data for each cancer type were randomly divided into a training set (80%) and a test set (20%). In the training set, five machine learning algorithms (Lasso, Random Forest, Boruta, SVM-RFE, and RFE) were applied to identify key genes, and intersection analysis was performed to generate a high-confidence candidate gene set. Based on this, ten machine learning algorithms (Lasso, GLM, Random Forest, SVM, XGBoost, Logistic Regression, KNN, LightGBM, Neural Network, and Decision Tree) were used to train predictive models. To further mitigate bias caused by overfitting, cross-validation (CV) AUC values were compared with test set AUC values to derive a comprehensive index—Model Score—which was subsequently used to select the optimal model for each cancer type (see Section 4) (Figure 4A; Appendix A). Based on the workflow described above, we first present the results of feature gene selection using the five machine learning algorithms in the training set, along with the distribution of candidate gene set sizes for each cancer type (Appendix A). Subsequently, the top five models based on Model Score values for each cancer were visualized (Figure 4B). The results demonstrated that the Model Scores of all optimal models exceeded 0.7, indicating that each cancer type achieved a predictive model with excellent performance (Figure 4C).

To systematically evaluate the performance of the optimal models, corresponding metastasis prediction models were established for each cancer type. Taking BLCA as an example, the model provides an intuitive prediction of individual metastasis risk. The calibration curve demonstrated a high concordance between predicted probabilities and actual outcomes, with an average error of 0.04, indicating high accuracy and robustness (Figure 4D). For other cancer types, the average error of the calibration curves was below 0.05, further supporting the reliability of the models (Appendix A). Based on the regression coefficients of the genes included in each model, Metastasis Prediction Score (MPS) was calculated for each cancer (Appendix A). Decision curve analysis (DCA) was then used to assess the potential clinical utility of the models. The results showed that the net benefit of each model across a wide range of threshold probabilities exceeded that of the “All intervention” (All) or “No intervention” (None) strategies, indicating high practical value and clinical applicability (Appendix A). To further validate the robustness of the models, additional metastatic cancer datasets from the GEO database were analyzed. The MPS scores of the metastasis groups were significantly higher than those of non-metastasis groups (Appendix A), and these findings were corroborated using scRNA-seq data (Appendix A). In addition, to assess whether these gene-level changes were reflected at the protein level, we downloaded pan-cancer proteomics datasets from the PDC database and observed that MPS scores were likewise significantly elevated in metastatic samples (Appendix A).

Through systematic analysis of metastasis signature genes, we found that, in addition to cancer-specific metastasis mechanisms, there are significant shared metastasis characteristics across different cancer types (Appendix A). This finding suggests the existence of a potential cross-cancer metastasis regulatory mechanism. Accordingly, genes that were up- or down-regulated in three or more cancers were selected as candidates, and all cancer datasets were combined for analysis. Using the same workflow as described previously (Appendix A), we constructed an optimal model with higher AUC in the overall test set (Appendix A; Appendix A). Although its performance in individual cancers was slightly lower than that of the cancer-specific models, the AUC of most cancers still exceeded 0.7, indicating a certain degree of generalizability (Figure 4E). Furthermore, we developed a global metastasis prediction model based on these key genes and validated its accuracy and clinical utility using calibration curves and decision curve analysis (DCA) (Appendix A). A Global Metastasis Prediction Score (GMPS) was established according to gene weighting coefficients (Appendix A). GMPS demonstrated strong predictive ability for metastasis across all cancer test sets (Figure 4F). In external bulk RNA-seq datasets, GMPS showed significant discriminative performance at the pan-cancer level. For individual cancer types, although its performance was slightly inferior to the cancer-specific Metastasis Prediction Score (MPS), GMPS still exhibited appreciable discriminative ability (Appendix A). Notably, GMPS was significantly elevated in the metastasis groups of TCGA Cervical Squamous Cell Carcinoma and Endocervical Adenocarcinoma (CESC) and Uterine Corpus Endometrial Carcinoma (UCEC) datasets, suggesting that the global model can not only assess metastasis risk at the multi-cancer level but also holds potential applicability in other epithelial cancers (Appendix A). Furthermore, GMPS also demonstrated consistent validation in both scRNA-seq and proteomics datasets (Appendix A).

To further investigate the relationship between tumor metastasis and the immune microenvironment, we analyzed immune cell infiltration. All immune cells in the single-cell dataset were re-clustered and annotated, resulting in 14 distinct subgroups (Figure 4G) based on canonical marker genes (Appendix A). Ultimately, these subgroups included two B cell subsets, dendritic cells (DCs), three macrophage subtypes, mast cells, monocytes, neutrophils, natural killer cells, plasma cells, and three T cell subsets. Subsequently, we examined the infiltration of various immune cell subtypes across different metastatic states. The results revealed that B cells and regulatory T (Treg) cells were significantly enriched in metastasis samples, whereas CD8+ T cells, CD4+ memory T cells, and DCs were predominantly enriched in non-metastasis samples (Figure 4H and Appendix A). This pattern suggests that a tumor microenvironment with high metastatic potential may suppress effector T cell activity by increasing the proportion of immunosuppressive Treg cells, thereby attenuating anti-tumor immune responses and facilitating metastasis [64]. In contrast, the enrichment of CD8+ and CD4+ memory T cells and DCs in non-metastasis samples indicates that effective anti-tumor immune surveillance may play a critical role in restraining metastasis. These findings are consistent with previously reported metastasis-associated immunosuppressive microenvironments, further supporting the central regulatory role of immune components in tumor metastasis [65,66,67].

In the RNA-seq data, samples were stratified into high-risk (High-MPS) and low-risk (Low-MPS) groups based on the median value of the MPS for each sample. The results showed that the proportion of high-risk samples was significantly higher in the metastasis group, whereas the proportion of low-risk samples was lower (Figure 4I), indicating that MPS can effectively discriminate samples with different metastasis risks and validating the reliability of the model in classifying metastasis status. Immune infiltration analysis was performed using the IOBR tool, revealing that B cells and Treg cells were significantly enriched in the metastasis group, whereas CD8+ T cells, CD4+ memory T cells, and DCs were relatively abundant in the non-metastasis group. These findings were consistent with the results of single-cell analysis (Figure 4J and Appendix A), further supporting the reliability of the metastasis prediction model in reflecting metastasis-associated immune microenvironment characteristics and providing a basis for understanding immune regulatory mechanisms in tumor metastasis. Notably, similar results were observed when the analysis was performed based on the GMPS (Appendix A).

Given the close correlation between the immune microenvironment and immunotherapy response, we further scored the high- and low-risk groups stratified by the MPS and GMPS using classical inhibitory immune checkpoints (IICs: PDCD1, CD274, CTLA4, LAG3, and TIGIT [68]) to assess the predictive ability of the models for immunotherapy response. The results showed that in BLCA, ccRCC, and GC, the IICs scores of the MPS high-risk groups were significantly higher than those of the low-risk groups, suggesting a strong association between the models and the sensitivity to immune checkpoint inhibitors (ICIs) in these cancers (Appendix A). Notably, at the pan-cancer level, the IICs scores of the GMPS high-risk group were also significantly elevated (Appendix A), further highlighting the potential clinical utility of the model in guiding treatment strategies.

In addition, to further evaluate the predictive value of MPS and GMPS for patient survival, samples were stratified into high- and low-score groups based on the optimal survival cut-off point, and Kaplan–Meier survival analysis was performed. Taking BLCA as an example, overall survival was significantly lower in the high-MPS group than in the low-MPS group (Log-rank, *p* < 0.001). The high-score group exhibited a rapid decline in survival during the early follow-up period, whereas the low-score group showed a relatively stable long-term survival trend (Figure 4K). At the pan-cancer level, patients in the high-GMPS group also demonstrated significantly poorer survival outcomes (Log-rank, *p* < 0.001) (Figure 4L). Particularly, in most cancer types, both high-MPS and high-GMPS groups were significantly associated with adverse clinical outcomes (Log-rank, *p* < 0.05) (Appendix A), indicating that the two models can not only effectively discriminate metastasis risk but also provide reliable prognostic information for patients.

### 2.5. Screening and Validation of Anti-Metastatic Drugs

The mechanism of cancer metastasis is highly complex, involving the coordinated action of multiple genes and their regulatory networks. In the previous analyses, we identified signature gene sets for different cancers and demonstrated their close association with metastasis. Building on this, we employed a drug screening framework that integrates Random Walk with Restart (RWR) and GSEA to identify potential small-molecule drugs capable of modulating metastasis-signature genes [69]. Specifically, we constructed a cancer-specific drug-target network by integrating drug targets obtained from the DrugBank database with cancer-specific metastasis-signature genes. Using these genes as seeds, we applied PageRank combined with the RWR algorithm to calculate a global drug score and identify candidate drugs for each cancer. To further assess the effects of these drugs, transcriptome data from drug-treated and control samples in the CMap database were subjected to differential expression and GSEA enrichment analyses, evaluating the regulatory impact of each drug on metastasis. Finally, the Drug Score derived from RWR and GSEA were integrated to generate a final drug ranking, yielding a prioritized list of metastasis-related candidate compounds (Figure 5A) (Appendix A).

We performed network visualization of the top ten drugs and their targets across various cancers, highlighting both unique and shared characteristics of drugs and targets among different cancer types (Figure 5B). Furthermore, the association between these drugs and metastasis, along with their specific Drug Score rankings, was systematically evaluated across all cancers (Figure 5C). The analysis revealed that Fostamatinib achieved the highest score in nine cancers, with most evidence supporting its anti-metastatic effects. Fostamatinib is an oral spleen tyrosine kinase (SYK) inhibitor reported to suppress breast carcinoma metastasis by targeting the SYK-driven EMT process [70]. Additionally, multiple studies indicate its potential anti-metastatic activity in esophageal carcinoma, head and neck squamous cell carcinoma, prostate carcinoma, and pancreatic carcinoma [71,72,73,74], suggesting a broad therapeutic potential for cross-cancer metastasis intervention.

On the other hand, we noticed that Eflornithine (also known as DFMO) showed a higher score in PRAD. DFMO has been shown in prostate cancer prevention studies to markedly suppress polyamine biosynthesis and inhibit tumor growth, indirectly suggesting a potential role in limiting metastatic progression. Notably, early-phase clinical trials further demonstrated that DFMO lowers intraprostatic polyamine levels and slows prostate tissue growth, reinforcing its promise as a chemopreventive agent [75,76]. In addition, Trifluoperazine has the most prominent score in HCC. Studies have shown that it can inhibit the migration and invasion of hepatocellular carcinoma cells and regulate metastasis-related pathways such as FOXO1 and VEGF [77]. The above results together prove that the drug screening model we used has good reliability, predictive ability and potential application value.

We further examined drug action patterns shared across multiple cancer types and identified a total of 22 drugs potentially associated with metastasis in two or more cancers (Appendix A). KEGG pathway enrichment analysis was performed on the top-ranked drug targets based on the Drug Score (Figure 5D). The results revealed that these targets were significantly enriched in several signaling pathways closely linked to cancer metastasis, including Focal adhesion, VEGF, Ras, and Wnt signaling pathways. Notably, the Focal adhesion pathway plays a critical role in regulating cell adhesion and migration [78], while the VEGF signaling pathway substantially influences tumor angiogenesis and metastatic progression [79]. Building on these findings, we conducted a drug repositioning analysis for multi-cancer metastasis, offering new strategies and insights for developing cross-cancer metastasis intervention approaches.

To assess the impact of candidate drugs on each cancer metastasis diagnosis model, we integrated gene expression data from drug-treated (Drug) and control (Ctrl) samples in the CMap database and applied the cancer-specific MPS weights to model both groups. The results showed that model scores in HNSCC and PRAD were significantly reduced following Fostamatinib treatment (Figure 5E), indicating that the drug can effectively modulate the expression of metastasis-related genes and further supporting its potential anti-metastatic effect in these cancers. Notably, in the PRAD metastasis model, most of the top-ranked drugs caused a marked decrease in MPS after treatment. This systematic response not only confirmed the sensitivity of the PRAD model to drug interventions but also reinforced the potential clinical utility of these drugs for prostate cancer metastasis management [74,80,81] (Figure 5E and Appendix A).

## 3. Discussion

In this study, we systematically investigated the dynamic evolution and intercellular interactions within the tumor microenvironment during cancer progression and metastasis by constructing a pan-cancer single-cell transcriptome atlas encompassing 12 common epithelial cancers and over 1.2 million cells. Our analyses not only recapitulated numerous well-established biological phenomena, but also uncovered a profound relationship between malignant epithelial cells and cancer-associated fibroblasts (CAFs) in terms of molecular characteristics and functional states. Importantly, these insights were further translated into predictive models and potential therapeutic targets with clinical relevance. The following sections summarize the main findings of this study.

First, our pan-cancer map analysis revealed that the proportion of malignant cells increased significantly with advancing TNM stage and metastatic progression, consistent with the theory of tumor clonal evolution, in which more invasive cells dominate and expand during disease progression [82]. Notably, fibroblasts, rather than malignant cells, exhibited the highest EMT scores. Cell–cell communication analysis further demonstrated that interactions between fibroblasts and malignant cells were the most frequent among all cell pairs and progressively intensified with metastatic progression. These interactions were highly enriched in extracellular matrix (ECM) remodeling pathways, including LAMININ and COLLAGEN signaling, suggesting that fibroblasts may facilitate tumor cell migration and invasion through ECM remodeling and play a pivotal role in metastasis [83].

Based on these findings, we focused on malignant cells and fibroblasts. By stratifying malignant cells according to EMT activity, we identified a subset of epithelial cells (hEMT) with high metastatic potential. This subgroup exhibits a characteristic ‘invasive’ phenotype, including elevated EMT activity, pronounced cancer stem cell (CSC) features [42,43,44,45], G1-phase cell cycle arrest consistent with the ‘go-or-grow’ hypothesis [46,47], and metabolic reprogramming, such as enhanced glycolysis [50,51]. CytoTRACE and Monocle analyses revealed a dynamic evolutionary trajectory from low- to high-score tumor cells, highlighting their plasticity and metastatic adaptability. Using non-negative matrix factorization (NMF), we identified 13 conserved biological programs across cancer types and found that hEMT cells were specifically enriched in stress response, immune regulation, and metastasis-related programs. Further, by integrating WGCNA with a mapping-based differential analysis strategy, we systematically identified two gene sets closely associated with metastasis. Together, these results not only provide a comprehensive molecular portrait of cells with high metastatic potential but also reveal key molecular characteristics underpinning tumor metastasis.

In-depth analysis of fibroblasts revealed that their metastasis-promoting function is primarily concentrated in the myCAF subtype. MyCAF not only exhibits pronounced EMT characteristics but also shows high activity in metastasis-related pathways, including WNT, hypoxia, and NOTCH signaling. Importantly, we observed a significant overlap between high-score epithelial cells and myCAF within the transcription factor regulatory network, including factors such as EGR2 and ETS1. EGR2 has been shown to promote metastasis by regulating cell migration and the immune status of the microenvironment [84], whereas ETS1 enhances cancer cell invasiveness by activating EMT- and matrix remodeling-related genes [85]. Moreover, integrated trajectory analysis using Slingshot and spatial transcriptomics (ST) suggested that a subset of highly malignant and plastic epithelial cells may further differentiate into the myCAF state through the EMT process, acquiring tumor-promoting functions. This finding deepens our understanding of the origin and plasticity of tumor microenvironment cells. Such continuous evolution of cell states may represent a novel mechanism for tumor immune evasion and distal colonization. Based on these trajectory dynamics, we further identified a set of genes closely associated with the metastatic process.

Based on the intersection of metastasis-associated characteristic genes identified in the mechanistic analyses above, we identified two metastasis prediction models, cancer-specific and global, which were constructed using a variety of machine learning methods. Both models demonstrated excellent predictive performance in independent cohorts. Analyses using nomograms and decision curves further supported their potential clinical utility for individualized metastasis risk assessment. The MPS and GMPS derived from these models also exhibited strong discriminative ability in external bulk RNA-seq datasets. Notably, GMPS maintained high predictive accuracy in cancer types not included in model training (CESC and UCEC), suggesting that it captures core features of cross-cancer metastasis. Importantly, MPS and GMPS were significantly associated with immune microenvironment characteristics: high-risk groups were enriched for immunosuppressive Treg cells, whereas low-risk groups were enriched for cytotoxic CD8+ T cells and DCs. These findings indicate that high metastatic risk is closely linked to an immunosuppressive tumor microenvironment, implying that our models can not only predict metastasis but also indirectly reflect patients’ immune status, potentially informing responses to immunotherapy. Survival analyses further confirmed that elevated MPS or GMPS levels were significantly associated with poor prognosis, supporting their role as independent prognostic risk factors.

Finally, based on a drug screening framework integrating random walk and GSEA, we systematically identified and validated potential anti-metastatic drugs across multiple cancers. The results demonstrated that this strategy exhibited robust predictive performance and strong biological rationale across different cancer types. Notably, Fostamatinib achieved the highest Drug Score across multiple cancers, suggesting its potential as a broad-spectrum anti-metastatic agent, consistent with previous studies demonstrating anti-metastatic effects in models of breast, esophageal, head and neck squamous cell, prostate, pancreatic, renal, colorectal, hepatocellular, and ovarian carcinomas [70,71,72,73,74,86,87,88]. Eflornithine and Trifluoperazine exhibited significant anti-metastatic effects in prostate cancer and hepatocellular carcinoma [75,76,77], indicating that this approach can simultaneously identify broad-spectrum as well as cancer-specific drugs. Mechanistic analyses revealed that shared drug targets were significantly enriched in classical metastasis-related pathways, including Focal adhesion, Wnt, Ras, and VEGF signaling, highlighting their critical roles in cell adhesion, migration, angiogenesis, and tumor microenvironment remodeling. Importantly, these pathways were recurrently enriched among multiple candidate drugs, suggesting common molecular targets for cross-cancer intervention in metastasis. Integrated analysis of MPS and CMap drug perturbation data revealed that Fostamatinib treatment significantly reduced MPS in HNSCC and PRAD, indicating its capacity to effectively suppress the overall expression of metastasis-related genes. Particularly, in PRAD, this effect exhibited a systematic trend, suggesting that the model can sensitively capture drug responses and providing a rationale for the optimization of potential future drug combinations.

Of course, this study has several limitations. First, although we integrated a large volume of public datasets, the sources are heterogeneous, and despite batch effect correction, residual biases may remain. Second, this study primarily relies on transcriptomic data, and validation at the protein level and through functional experiments would further strengthen our conclusions. For instance, the hypothesis that malignant epithelial cells can transdifferentiate into myCAF requires confirmation via in vivo approaches, such as lineage tracing. Third, our predictive models still require further validation and optimization in prospective clinical cohorts. Fourth, given that this study specifically focuses on EMT, the applicability of our findings may be largely limited to epithelial cancers. In addition, the clinical efficacy of the identified candidate drugs needs to be evaluated in future studies. These studies should also include in vitro and in vivo experiments to validate the anti-metastatic potential of the top candidate compounds, thereby strengthening the translational impact of our computational predictions.

In summary, through systematic multi-omics analyses, this study uncovers a novel mechanism by which malignant epithelial cells and fibroblasts drive metastasis via dynamic interactions within the tumor microenvironment. Moreover, the metastasis-signature genes identified were translated into prediction models and candidate therapeutic agents, offering new strategies for prognosis prediction and the precision prevention and treatment of cancer metastasis.

## 4. Materials and Methods

### 4.1. scRNA-Seq Data Acquisition and Inclusion Criteria

We systematically searched for and retrieved single-cell RNA sequencing (scRNA-seq) datasets from multiple publicly available repositories, including GEO (Gene Expression Omnibus, https://www.ncbi.nlm.nih.gov/geo/, accessed on 15 July 2024), OMIX (https://ngdc.cncb.ac.cn/omix/, accessed on 9 August 2024), ArrayExpress (https://www.ebi.ac.uk/biostudies/arrayexpress/, accessed on 2 September 2024), Single Cell Portal (https://singlecell.broadinstitute.org/single_cell, accessed on 15 September 2024), and SRA (https://trace.ncbi.nlm.nih.gov/Traces/sra/, accessed on 18 October 2024). Datasets were included if they met the following criteria: (i) derived from primary tumor samples, and (ii) clinical information on cancer stage or metastasis status was explicitly provided by the study authors.In total, 32 scRNA-seq datasets comprising 265 tumor samples were collected and integrated. The samples included 37 stage I, 65 stage II, 91 stage III, and 72 stage IV tumors. Based on metastasis status, samples were further categorized into 108 non-metastasis (NM), 74 with regional metastasis (RM), 49 with distant metastasis (DM), and 34 with unknown metastasis (UM) (Appendix A).

### 4.2. scRNA-Seq Data Preprocessing

We first processed and integrated scRNA-seq data from each cancer type separately, generating 12 Seurat objects. For each object, we applied the Seurat package (v5.1.0) [89] to perform quality control (QC), normalization, and highly variable gene (HVG) selection. QC thresholds were defined according to the characteristics of each dataset: cells with low gene counts or a high proportion of mitochondrial transcripts were removed, and only high-quality cells were retained for downstream analysis. We then merged data layers using the JoinLayers function and applied ScaleData to normalize and scale gene expression. Principal component analysis (PCA) was performed on HVGs, and the number of informative dimensions was determined by the ElbowPlot. To correct for batch effects, Harmony (v1.2.1) [90] was applied, and nearest-neighbor graphs were constructed using the corrected embedding space for clustering analysis. Uniform Manifold Approximation and Projection (UMAP) was used for visualization. Cell types were annotated by integrating canonical markers from literature and CellMarker 2.0 database [91]. We further identified cluster-specific marker genes using the FindAllMarkers function and visualized their expression profiles and functional features with ClusterGVis (v0.1.2).

### 4.3. Identification of Malignant Epithelial Cells

To distinguish malignant from non-malignant epithelial cells, we applied a two-step strategy based on the CNV matrix generated by inferCNV (v1.20.0) [92], adapted from a previously published approach for malignant cell identification [93]. First, we obtained the CNV Scores from inferCNV. Based on these values, we calculated a CNV Deviation for each cell, which quantifies the overall deviation of its copy number profile from the neutral state. The calculation was performed as follows:(1)$${\mathrm{CNV}\;\mathrm{Deviation}}_{j}\;=\;\frac{1}{n}\sum_{i=1}^{n}{{(\mathrm{CNV}Score}_{i,j}-1)}^{2}$$

Here, ${\mathrm{CNV}Deviation}_{j}$ represents the CNV Deviation of the j-th cell, ${\mathrm{CNV}Score}_{i,j}$ denotes the CNV Score of the i-th gene in the j-th cell, and n is the total number of genes.

Second, to further distinguish malignant cells from non-malignant ones, we implemented a correlation-based approach. We selected the top 5% of cells with the highest CNV Deviation from the tumor-derived cells and calculated the average CNV Deviation across these cells to serve as a reference CNV curve. We then computed the Pearson correlation coefficient between each cell’s CNV Deviation and this reference curve. Finally, we integrated the CNV Deviation and correlation information, using histograms to determine threshold values. Cells with CNV Deviation > 0.002 and correlation > 0.1 were labeled as Malignant, cells with CNV Deviation < 0.002 and correlation < 0.1 were labeled as Not Malignant, and the remaining cells were designated as Other. Finally, only epithelial cells identified as malignant were retained, and a new Seurat object was constructed for subsequent analysis.

### 4.4. Gene Set Signature Scores

We obtained the EMT-related gene set from the GSEA database (https://www.gsea-msigdb.org/gsea/msigdb/cards/HALLMARK_EPITHELIAL_MESENCHYMAL_TRANSITION.html, accessed on 18 November 2024) and applied the AddModuleScore function (Seurat) to calculate an EMT score for each cell based on the expression of EMT-associated genes. To enable cross-cell comparisons, the scores were normalized and scaled to a range of 0–1. Similarly, the scoring of CSC-related genes and others in the study was performed using the same method, with the CSC gene set obtained from [94].

### 4.5. Cell–Cell Interaction Analysis

To investigate interactions among different cell types, we employed the CellChat package (v1.6.1) [95] to analyze intercellular communication based on single-cell transcriptomic data. The expression matrix was first processed using the human CellChat database (CellChatDB.human), which involved filtering signaling-related genes, identifying highly expressed genes and ligand-receptor pairs, and mapping them onto the protein–protein interaction (PPI) network. Communication probabilities between cells were then calculated using the computeCommunProb function, and low-abundance cell populations with unreliable signals were removed. Probabilities were further aggregated at the signaling pathway level using computeCommunProbPathway to construct a global communication network. Finally, both the number and strength of interactions across cell populations were visualized using netVisual_bubble and netVisual_circle, enabling a comprehensive assessment of intercellular communication patterns.

### 4.6. Cell Cycle Analysis

To assess the impact of cell cycle status on different cell populations, we first extracted Seurat objects corresponding to three EMT subtypes. Using the built-in cell cycle gene sets for S phase and G2/M phase in the Seurat package, we calculated the S phase score (S.Score) and G2/M phase score (G2M.Score) for each cell with the CellCycleScoring function, which also allowed inference of the cell cycle stage (Phase). Finally, we visualized and compared the cell cycle distributions across the EMT subtypes to highlight differences in cell cycle states among them.

### 4.7. GEVA Enrichment Analysis

We employed the GSVA package (v1.52.3) [96] to assess pathway activity and used the msigdbr package (v7.5.1) to obtain human Hallmark (H), KEGG (C2: KEGG), and GO biological process (C5: BP) gene sets. For each cancer type, cells were extracted from the corresponding Seurat object, and an average expression matrix was computed for each cell type or cluster to reduce noise and represent the typical transcriptional profile. GSVA was then applied to each gene set using the default non-parametric kernel estimation method, producing pathway enrichment scores for each cell type. Subsequently, Z-score normalization was performed across cells within each cancer type to enable cross-cancer comparisons of pathway activity. Enrichment patterns were visualized using heatmaps to highlight subtype-specific pathway regulation.

### 4.8. Cell Differentiation and Trajectory Analysis

We integrated CytoTRACE (v0.3.3) [97] and Monocle (v2.32.0) [98] to evaluate the differentiation status and pseudotemporal relationships of EMT subtype cells. This method was derived from a pan-cancer study [99]. For each cancer type, a CellDataSet was constructed and preprocessed in Monocle, including size factor estimation, dispersion calculation, and gene detection. Differential gene analysis was performed using Seurat-identified highly variable genes, and genes with q-values < 0.01 were selected as input for DDRTree dimensionality reduction and orderCells, generating pseudotime trajectories for each cancer. CytoTRACE was then applied to compute the differentiation potential of malignant cells, with lower scores corresponding to higher differentiation potential. Dimensionality reduction coordinates, pseudotime, cell states, and cell type annotations were extracted from Monocle objects to construct phenotypic vectors. The PlotCytoTRACE function was used to map differentiation potential onto the reduced dimensional space and visualize it by cell type. Finally, we analyzed the correlation between differentiation potential and EMT subtypes to assess its potential role in tumor metastasis.

### 4.9. Non-Negative Matrix Factorization (NMF)

To systematically identify cross-sample robust gene modules in malignant cells for each cancer type, we applied non-negative matrix factorization (NMF, v0.27) [100]. The integrated Seurat object was first split by sample, retaining only those with more than 5 cells. Each sample was independently normalized using SCTransform, and highly variable genes were selected as the NMF input matrix. Decomposition was repeated 10 times for each rank (from 4 to 12), modules were extracted based on gene weights, and the rank with the optimal composite metric was chosen to obtain the final module set for each sample. Cross-sample integration was then performed by counting the recurrence frequency of genes to select stable genes. Modules with Jaccard similarity > 0.05 to at least five other modules were retained as robust. A gene co-occurrence network was constructed (genes as nodes, co-occurrence frequency as edge weights), high-connectivity genes were retained, and the network was partitioned using the Infomap algorithm to define cross-sample robust gene modules. Modules with similar structures across cancers were merged, resulting in 6–7 modules per cancer and a total of 13 core modules. Functional annotation was performed using ClusterProfiler (v4.11.0) [101] for HALLMARK and GO biological process (BP) terms, and modules were defined as distinct ‘functional programs’ based on characteristic genes. Module activity scores for each cell were calculated using AUCell (v1.24.0), Z-score normalized within each cancer type for cross-cancer comparisons, and differences among EMT subtypes were assessed using the Wilcoxon test.

### 4.10. Gene Co-Expression Module Identification

To systematically analyze the functional module characteristics of hEMT cells across different cancers, we independently constructed a weighted gene co-expression network within each cancer type to minimize bias from inter-cancer differences. Using the hdWGCNA (v0.2.19) [102] package, hEMT cells were first aggregated into metacells with the MetacellsByGroups function. Gene-gene expression correlations were then calculated, and a signed co-expression network was constructed using a soft-thresholding approach. Gene modules were subsequently identified based on the topological overlap matrix (TOM). Previously defined AUCell activity scores of the NMF functional modules were treated as phenotypic features, and their correlations with hdWGCNA module eigengenes (MEs) were computed and tested for significance. Finally, modules significantly associated with metastasis-related functional programs were selected for each cancer type using the thresholds |r| > 0.2 and *p* < 0.05, and their gene sets were extracted to form the first batch of metastasis-related gene sets (GeneSet1).

### 4.11. Mapping-Based Differential Gene Analysis

To identify differential genes more accurately, we developed a mapping-based differential expression analysis approach. We first re-clustered hEMT and lEMT cells for each cancer type. Standard single-cell RNA-seq preprocessing was applied, including LogNormalize normalization, identification of highly variable genes, data scaling, and PCA for dimensionality reduction. The first 30 principal components were used to build a nearest-neighbor graph for clustering, followed by t-SNE visualization. To systematically assess the correspondence between hEMT and lEMT cells, we employed Seurat’s anchor mapping approach (FindTransferAnchors/TransferData) to perform bidirectional mapping, projecting hEMT cells onto lEMT cells and vice versa to identify corresponding cell pairs. Highly similar cells were then removed to reduce shared background interference, and differential expression analysis was performed on the remaining populations using the FindMarkers function. By applying a significance threshold (*p* < 0.05, |avg-log2FC| > 0.25), we obtained the second batch of metastasis-related feature gene sets (GeneSet2).

### 4.12. pySCENIC Analysis

We applied a single-cell regulatory network inference and clustering workflow (pySCENIC, v0.12.1) [103] to evaluate the activity and characteristics of transcription factors in three EMT epithelial subtypes and three CAF subtypes. First, GRNBoost2 was used to infer co-expression relationships between transcription factors and candidate target genes, constructing an initial regulatory network. The human genome hg38 served as a reference, and motif enrichment was performed using motifs v10 nr.hgnc, retaining only high-confidence regulatory pairs. AUCell was then used to calculate regulon AUC scores for each cell, and the Regulon Specificity Score (RSS) was computed for each transcription factor. To identify shared regulatory factors across cancers, the top 30 RSS-ranked factors from each cancer were selected, and transcription factors present in at least four cancer types were integrated for visualization.

### 4.13. Slingshot Trajectory Analysis

Based on single-cell transcriptome data of hEMT and myCAF cells across different cancers, pseudotime trajectories were constructed using Slingshot (v2.12.0) [104], with hEMT cells as the starting point after UMAP dimensionality reduction. Trajectories were visualized in UMAP space, and the dynamic distribution of different cell types along the trajectory was assessed by density analysis. The expression matrix was then converted into a Monocle CellDataSet, and differential expression analysis was performed based on cell type (q < 0.001). Significantly differentially expressed genes were used for pseudotime ordering, and dynamic expression curves were calculated using genSmoothCurves. Mann–Kendall tests were applied to identify genes with significant upward or downward trends along pseudotime (*p* < 0.05, |τ| > 0.7), forming the third batch of transfer-related gene sets (GeneSet3). Finally, GO enrichment analysis of up- and downregulated genes was performed using ClusterProfiler, and the top ten pathways in Biological Process (BP), Cellular Component (CC), and Molecular Function (MF) were visualized.

### 4.14. Spatial Transcriptome (ST) Data Collection and Trajectory Analysis

We collected spatial transcriptome (ST) data for three cancers from the GEO database (Appendix A), and analyzed them using Scanpy (v1.9.3) [105] and stlearn (v0.4.12) [106]. Genes expressed in fewer than three spatial spots were filtered out, and the data were normalized, log-transformed, and scaled. To extract high-order spatial features from tissue sections, tissue images were tiled using st.pp.tiling, and deep-learning–based morphological features were extracted via st.pp.extract_feature, which applies a pretrained convolutional neural network to each tile. Based on these features, the SME (Spatial Morphological Expression) method was applied to spatially smooth and standardize the expression matrix, generating a spatially optimized expression matrix. PCA was then performed for dimensionality reduction, a neighbor graph was constructed, and Louvain spatial clustering was applied. CARD (v1.1) [107] was used to deconvolute ST capture sites and predict major cell types, using the previously annotated single-cell transcriptome data as reference. Spatial expression differences were evaluated by integrating morphological and transcriptomic features. Clusters corresponding to high-expression regions of hEMT and myCAF cells were extracted, and the trajectories of the two selected clusters were analyzed and visualized using the st.pl.cluster_plot function (stlearn).

### 4.15. Metastasis-Signature Gene Sets

Based on the three predefined GeneSets, we identified 12 metastasis-signature gene sets by intersecting results across different cancer types. Using the up- and down-regulation information from GeneSet2, which was derived from the direct comparison between hEMT and lEMT cells and thus provides a robust phenotypic reference, all genes were directionally annotated, resulting in two groups of EMT-associated genes (upregulated and downregulated) for each cancer type. To further investigate their biological functions, we selected genes consistently up- or downregulated in at least two cancers and performed GO enrichment analysis (ontology: BP) with the ClusterProfiler package to identify significant pathways (*p* < 0.05). To systematically illustrate interrelationships among pathways, we employed the aPEAR package (v1.0.0) [108] to construct an enriched pathway network, calculated pathway similarity using the Jaccard coefficient, and applied hierarchical clustering to group related pathways. Finally, clusters containing at least six pathways were retained for visualization, and the network was plotted using the plotPathClustersExpanded function.

### 4.16. Bulk RNA-Seq Data Acquisition and Preprocessing

RNA-seq data and corresponding clinical information for 12 cancer types were obtained from the TCGA (The Cancer Genome Atlas, https://www.cancer.gov/ccg/research/genome-sequencing/tcga, accessed on 25 February 2025) and GEO databases, consistent with those used in the single-cell analysis. The datasets included BRCA, BLCA, KIRC, COAD, READ, ESCA, STAD, LIHC, HNSC, LUSC, LUAD, OV, PAAD, and PRAD, with COAD and READ merged as CRC and LUSC and LUAD merged as LC. During preprocessing, samples lacking metastasis information were excluded, and the remaining ones were classified into non-metastasis and metastasis groups based on TNM stage. Genes with zero variance and samples or genes containing missing or infinite values were removed. Expression matrices were then normalized, and only genes expressed in more than 10% of samples were retained. To correct for batch effects across datasets, the ComBat function from the sva package (v1.1) [109] was applied. In total, 2992 samples were obtained, including 1407 non-metastasis and 1585 metastasis cases, which were used for subsequent analyses (Appendix A).

### 4.17. Feature Extraction and Best Model Selection in Machine Learning

To ensure the reliability and generalizability of model training, the data were randomly split into an 80% training set and a 20% test set. Using the metastasis-signature genes, five machine learning methods (Lasso, Random Forest, Boruta, SVM-RFE, and RFE) were applied to further identify key genes in the training set, with metastasis status as the dependent variable and the gene expression matrix as the independent variables. Lasso was built using the glmnet package (v4.1-8) [110], with the optimal λ determined by 10-fold cross-validation and nonzero coefficients at λ.min retained as features. Random Forest was constructed using the randomForest package (v4.7-1.1) [111] with 10,001 trees, and the top 30–40 genes ranked by MeanDecreaseGini were selected. Boruta was implemented with the Boruta package (v9.0.0) [112] under pvalue = 0.001 and maxRuns = 20, retaining “Confirmed” and “Tentative” genes. SVM-RFE was performed using e1071 (v1.7-14) [113] with 10-fold cross-validation and a linear kernel, iteratively removing lowest-weight genes and selecting the top 30–40. RFE was conducted using the caret package (v6.0-94) [114] with bootstrap-resampled recursive feature elimination to identify optimal features. Intersection analysis across these methods generated high-confidence candidate gene sets, which were then used to train ten machine learning models—Lasso [110], GLM, RF [111], SVM, XGBoost [115], Logistic Regression, KNN, LightGBM [116], Neural Network [117], and Decision Tree. All models were evaluated in the training set and independently validated in the test set. To reduce the risk of overfitting due to a small sample size, we combined the area under the curve (AUC) from cross-validation (CV) with the AUC from the test set, and calculated a comprehensive Model Score through weighted fitting. This score was then used to select the best model for each cancer type.(2)$$\mathrm{Model}\;\mathrm{Score}=(\frac{\mathrm{CV}+\mathrm{Test}}{2})\;-\lambda \;\times \;\left|\;\mathrm{CV}\;-\;\mathrm{Test}\;\right|$$

The weight coefficient λ was set to 0.8.

### 4.18. Construction and Validation of Metastasis Prediction Model

To linearize the best model, multi-gene logistic regression models were constructed with the rms package (Version 6.3-0) using the lrm function to fit metastasis status, and regression coefficients were used to calculate the metastasis prediction score. Nomograms were generated using regplot to illustrate individual sample predictions and each gene’s contribution, and calibration was performed using the bootstrap method (B = 1000) via the calibrate function. Decision curve analysis (DCA) was conducted with the rmda package (Version 1.6) to compare the clinical net benefits of the multi-gene model and assess its predictive performance, and regression coefficients were extracted to calculate the metastasis prediction score (MPS/GMPS). Moreover, we further validated the MPS/GMPS in independent pan-cancer datasets, including RNA-seq cohorts from the GEO and TCGA databases, as well as proteomics datasets from the PDC database (https://pdc.cancer.gov/pdc/browse, accessed on 8 April 2025) (Appendix A).

### 4.19. Immune Infiltration Analysis

First, MPS and GMPS scores were calculated for each sample. Based on these scores, samples were divided into high-risk and low-risk groups according to the median. The distribution of samples across different metastasis statuses was then compared between the high- and low-risk groups. The deconvo_tme function in the IOBR package (Version 0.99.0) [118], combined with the CIBERSORT algorithm using the LM22 signature matrix, was used to estimate the proportion of immune cell infiltration in each sample. Subsequently, the average proportions of each immune cell type in the high- and low-risk groups across different cancers were visualized.

### 4.20. Survival Analysis

RNA-seq data for each cancer were integrated with clinical follow-up information (time and event). Based on MPS and GMPS, the surv_cutpoint function from the survminer package (v0.4.9) was used to determine the optimal cutoff, dividing samples into high- and low-expression groups. Kaplan–Meier survival curves were constructed using survfit, and differences between groups were assessed with the log-rank test. Curves were visualized with ggsurvplot, showing survival probability over time, along with risk tables and censored event markers. The confidence interval was set at 95%.

### 4.21. Screening of Metastasis-Related Drugs

To further identify potential drugs targeting metastasis-signature gene sets, we employed a series of drug prioritization methods [69]. We obtained drug and target information from the DrugBank database (https://go.drugbank.com/, accessed on 5 June 2025), combined with cancer-specific metastasis-signature genes, and integrated the STRING protein–protein interaction (PPI) network to construct a cancer-specific drug-target network. Using metastasis-signature genes as seeds, initial scores were assigned to nodes via the PageRank algorithm. Based on the Random Walk with Restart (RWR) algorithm [119], global scores for drugs within the network were calculated, and potential candidate drugs were independently identified for each cancer type. The RWR algorithm is based on the following formula:(3)$$P_{t+1\;}=\;\alpha AP_{t}\;+\;(1-\alpha ){\;P}_{0}$$

Here, $P_{0\;}$ represents the initial binary probability vector, A denotes the adjacency matrix of the drug-target network, α determines the extent of network diffusion. We used the optimal value (α = 0.7) in our analysis.

Furthermore, we used the Connectivity Map (CMap) database (https://clue.io/, accessed on 21 June 2025) [120] to obtain gene expression profiles under drug-treated and control conditions in cancers. Differential expression analysis was performed using the limma package (v0.4.9) [121], and GSEA enrichment analysis was conducted based on a predefined set of metastasis-signature genes. Normalized enrichment scores (NES) were calculated to determine whether a drug inhibits or promotes the expression of metastasis-associated genes. Finally, we took the intersection of drugs identified by both methods and combined the RWR scores with the GSEA results in a weighted manner to generate a comprehensive score for each drug.(4)$$\mathrm{Drug}\;\mathrm{Score}=\lambda_{1}\;\times {\;\mathrm{Score}}_{\mathrm{RWR}\;}+\lambda_{2\;}\times {\;\mathrm{Score}}_{\mathrm{GSEA}}$$

Here, ${\mathrm{Score}}_{\mathrm{RWR}\;}$ is the drug score from the random walk, ${\mathrm{Score}}_{\mathrm{GSEA}}$ is the drug score from the GSEA enrichment analysis, and the weights $\lambda_{1}$ and $\lambda_{2}$ are 0.6 and 0.4, respectively.

### 4.22. Validation of Metastasis-Related Drugs

For the top-ranked drugs shared across multiple cancers, their target genes were extracted from the DrugBank database, and KEGG enrichment analysis was performed using ClusterProfiler to validate the reliability of the drug screening model. Subsequently, using gene expression data from drug-treated (Drug) and control (Ctrl) groups in the CMap database, the distribution of MPS across candidate drugs was evaluated. Finally, differences in scores between the two groups were compared using the Wilcoxon rank-sum test to assess statistical significance.

## 5. Conclusions

In this study, we systematically analyzed the dynamic characteristics of EMT within the tumor microenvironment and their association with metastasis by constructing a pan-cancer single-cell atlas. The metastasis prediction systems developed, MPS and GMPS, not only demonstrated excellent cancer specificity and generalizability, but also provided robust tools for clinical metastasis risk assessment, with additional potential for evaluating immune status and patient prognosis. By integrating multi-omics analyses with a drug repositioning strategy, we identified candidate broad-spectrum anti-metastatic agents, exemplified by Fostamatinib. These findings not only advance our understanding of the molecular mechanisms underlying tumor metastasis, but also offer novel biomarkers and potential therapeutic strategies for accurate prediction, prognostic evaluation, and targeted intervention in cancer metastasis.

## Figures and Tables

**Figure 1 ijms-26-11582-f001:**
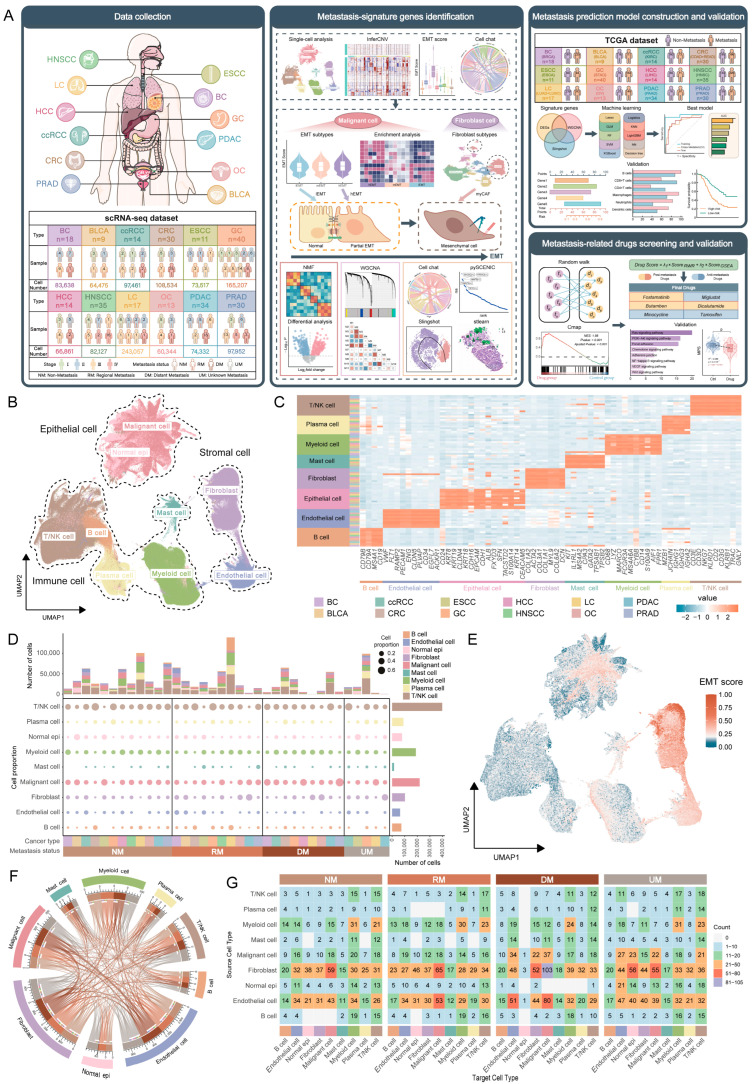
Single-cell transcriptomic landscape across 12 epithelial cancers reveals metastasis-associated tumor microenvironment. (**A**) Overview of the study workflow outlining four components: data collection, metastasis-signature gene identification, metastasis prediction model construction and validation, and metastasis-related drug screening and validation. (**B**) UMAP visualization of 1,217,505 cells integrated across 12 cancer types, annotated into eight major cell populations. (**C**) Heatmap of canonical marker gene expression for the cell types defined in (**B**), with color intensity representing average expression levels. (**D**) Comparative composition of cell type proportions across metastatic states in 12 cancer types. Dot plots depict relative proportions, showing a pronounced increase of malignant cells with advancing metastatic status. (**E**) UMAP visualization of the distribution of signature scores based on 200 EMT genes across all cells, highlighting fibroblasts and malignant cells as the major EMT-enriched populations. (**F**) Chord diagram illustrating cell–cell interactions among cell types across different metastatic states. Fibroblast–malignant cell interactions are the most prominent across all states. (**G**) Heatmap of interaction frequencies demonstrating that fibroblast–malignant cell interactions intensify with progressive metastasis.

**Figure 2 ijms-26-11582-f002:**
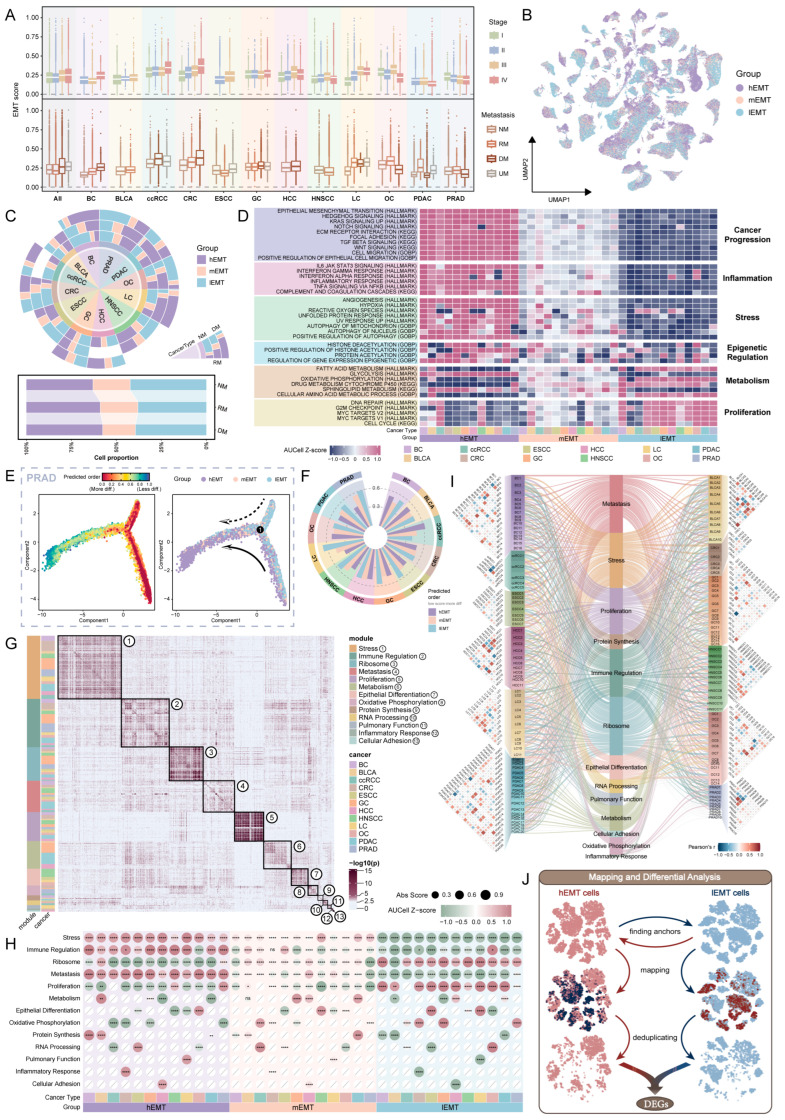
Functional heterogeneity of malignant epithelial subpopulations and characterization of their metastatic features. (**A**) Boxplots showing EMT signature scores in malignant cells across cancer stages and metastatic states, demonstrating that EMT activity progressively increases with tumor progression and metastasis. (**B**) UMAP visualization of malignant cells showing three EMT-defined subtypes (hEMT, mEMT, lEMT). (**C**) Stacked bar plot showing that the proportion of hEMT cells increases with metastatic advancement across cancer types. (**D**) Heatmap illustrating AUCell Z-score activity of functional pathways across EMT subtypes, with pathways hierarchically clustered by biological functions, highlighting distinct enrichment patterns among different EMT subtypes. (**E**) Developmental trajectories of PRAD cells: CytoTRACE (**left**) shows predicted differentiation potential (red = low, blue = high), and the Monocle trajectory (**right**) depicts lineage progression with annotated branch points, revealing a pseudo-temporal transition from lEMT to hEMT states. Different arrows indicate distinct developmental trajectories, and numbers mark key branch points. (**F**) Circular bar plot of differentiation potential scores for EMT subgroups (lower scores indicate higher differentiation potential), showing that hEMT cells possess greater stem-like plasticity. (**G**) Heatmap summarizing 2004 cancer-derived modules, integrated into 13 major programs using the Infomap algorithm. (**H**) Dot plot showing AUCell scores of 13 NMF-defined programs across EMT subgroups in different cancer types, highlighting the preferential functional programs of each EMT subtype. Statistical significance was assessed using the Wilcoxon rank-sum test (* *p* < 0.05, ** *p* < 0.01, *** *p* < 0.001, **** *p* < 0.0001, ns = not significant). (**I**) Heatmap (**left** and **right**) depicting WGCNA module correlations, with the Sankey diagram (**middle**) showing associations between WGCNA modules and NMF functional program scores. Ribbon width indicates correlation strength, and gray ribbons represent non-significant associations (*p* > 0.05 or |cor| < 0.1). (**J**) Schematic workflow of the mapping-based differential gene analysis strategy.

**Figure 3 ijms-26-11582-f003:**
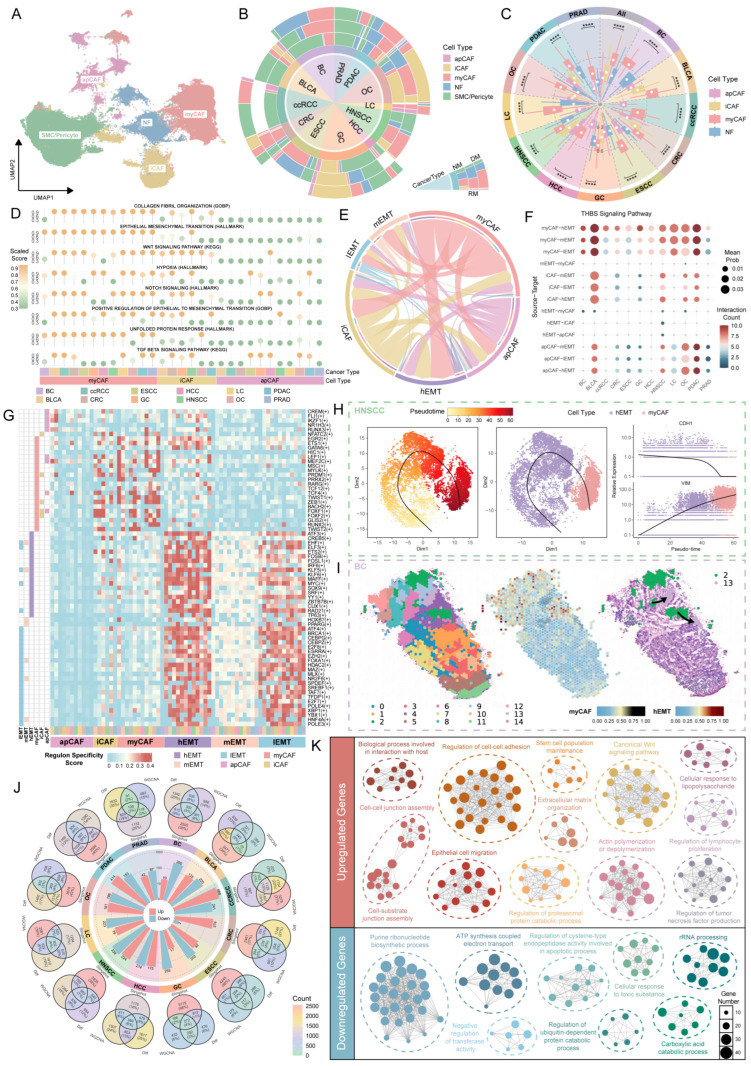
Fibroblast subtype characteristics and their dynamic interactions with malignant epithelial cells. (**A**) UMAP visualization of fibroblast subtypes annotated based on canonical marker genes, and classified into five distinct cell types. (**B**) Circular stacked bar plot showing the proportions of fibroblast subtypes across metastatic states in different cancers, highlighting the expansion of iCAFs and myCAFs with tumor progression. (**C**) Circular boxplots depicting EMT gene scores across fibroblast subtypes, showing myCAFs as the major EMT-enriched fibroblast population, with statistical significance indicated (**** *p* < 0.0001). (**D**) Lollipop plots illustrating pathway enrichment scores across eight metastasis-related pathways, revealing preferential enrichment of myCAFs in pathways promoting invasion and metastasis. (**E**) Chord diagram of intercellular interactions (excluding self- and intra-type interactions), indicating strong communication between myCAFs and hEMT cells. (**F**) Dot plot showing intercellular interactions within the THBS signaling pathway across different cancers (excluding self- and intra-type interactions). (**G**) Heatmap of regulatory activity of key transcription factors across EMT subtypes and CAF subtypes, with bars indicating subtype origin, revealing convergence of transcriptional programs between myCAFs and hEMT cells. (**H**) Slingshot trajectory of HNSCC fibroblasts, showing pseudotime progression (**left**), subtype assignment (**middle**), and dynamic expression of CDH1 and VIM (**right**), indicating a potential transition from hEMT cells toward myCAFs. (**I**) Leiden clustering of BC into subclusters 0–14 (**left**), spatial expression patterns of hEMT cells (black circles) and myCAFs (red squares) (**middle**), and the inferred spatial differentiation trajectory from hEMT cells to myCAFs (**right**). (**J**) Integrative analysis using Diff, WGCNA, and Slingshot across cancers to identify core gene sets, with the central circular heatmap showing cancer-specific metastasis-signature genes and their regulation direction and counts (red = up, blue = down), and peripheral Venn diagrams depicting method overlaps. (**K**) Network visualization of representative GO terms and pathways enriched in metastasis-related genes commonly upregulated (**top**) or downregulated (**bottom**) across two or more cancer types, illustrating the functional programs of these genes. Nodes represent significantly enriched GO terms (*p* < 0.05), and edges indicate similarity between pathway-associated gene sets.

**Figure 4 ijms-26-11582-f004:**
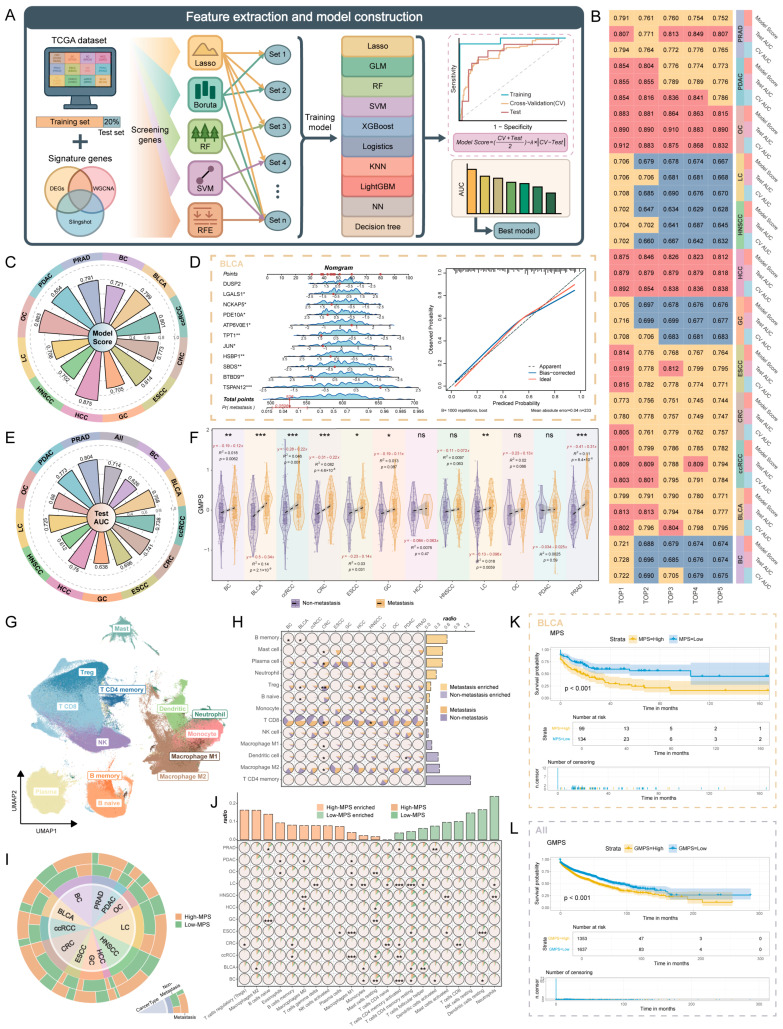
Construction and validation of metastasis prediction models and analysis of their immunological and prognostic implications. (**A**) Workflow for feature extraction and model construction, integrating multiple machine learning algorithms to select high-confidence metastasis-signature genes. (**B**) Heatmap showing the CV AUC, Test AUC, and Model Score of the top 5 models in each cancer type. (**C**) Circular bar plot depicting the Model Scores of the cancer-specific best models, demonstrating high prediction accuracy (Model Score > 0.7) across cancers. (**D**) Nomogram of the BLCA metastasis prediction model (**left**), showing individual gene scores and total score corresponding to predicted metastasis risk. Statistical significance was assessed using the Wilcoxon rank-sum test (* *p* < 0.05, ** *p* < 0.01, *** *p* < 0.001). The right panel displays the calibration curve, with predicted versus observed metastasis probability. (**E**) Circular bar plot showing Test AUC of the pan-cancer best model. (**F**) Boxplots showing GMPS distributions with metastatic samples exhibiting significantly higher scores, confirming the global predictive ability. Statistical significance was assessed using the Wilcoxon rank-sum test (* *p* < 0.05, ** *p* < 0.01, *** *p* < 0.001, ns = not significant). (**G**) UMAP plot of annotated immune cell types in single-cell data, identifying 14 distinct subgroups. (**H**) Immune cell infiltration in single-cell data under different metastasis statuses. Left: heatmap showing differences and significance in immune cell proportions (* *p* < 0.05, ** *p* < 0.01). Right: bar plot of relative changes per cell type, colored by group enrichment. (**I**) Circular bar plot showing proportions of High-MPS and Low-MPS samples per cancer type, showing high-risk samples are enriched in metastasis cases. (**J**) Immune infiltration landscape of different MPS risk groups in RNA-seq data. Bottom: heatmap showing differences and significance in immune cell proportions (* *p* < 0.05, ** *p* < 0.01, *** *p* < 0.001). (**K**,**L**) Association of MPS with survival in BLCA (**K**) and GMPS with survival in pan-cancer (**L**). Kaplan–Meier curves depict survival of high- and low-score groups over time (months), with at-risk and censored patients indicated. Shaded areas represent confidence intervals. High-score groups exhibit significantly worse survival (*p* < 0.001).

**Figure 5 ijms-26-11582-f005:**
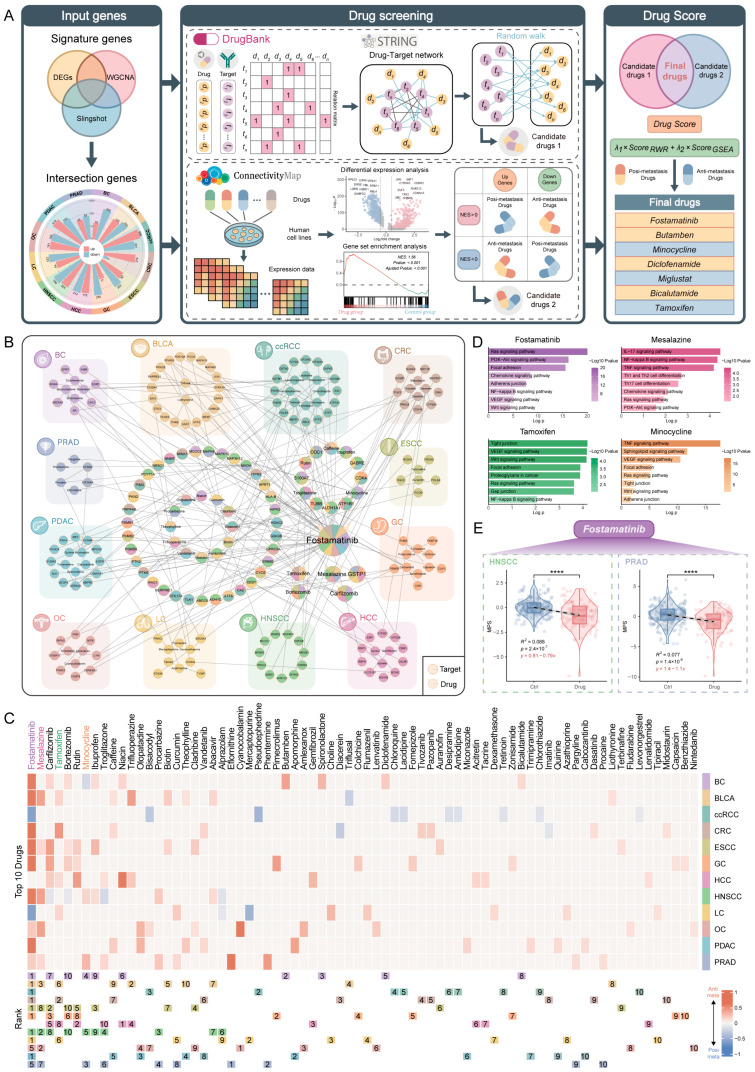
Screening and validation of anti-metastatic drugs. (**A**) Schematic diagram of the metastasis-related drug screening model, illustrating the overall workflow from input data to drug screening model construction and subsequent drug scoring. (**B**) Network diagram showing the top 10 ranked drugs and their corresponding targets in each cancer type, highlighting both shared and cancer-specific drug-target relationships. (**C**) Heatmap of the top 10 candidate drugs across cancers, showing Drug Score (color intensity), regulatory effect on metastasis (red: inhibition; blue: promotion), and overall drug ranking; Fostamatinib ranks highest in nine cancers, indicating broad anti-metastatic potential. (**D**) Bar plots showing KEGG pathway enrichment of targets for Fostamatinib, Mesalazine, Tamoxifen, and Minocycline, highlighting significant enrichment in metastasis-related pathways. (**E**) Boxplots comparing MPS between drug-treated (Drug) and control (Ctrl) groups in HNSCC and PRAD, demonstrating that Fostamatinib treatment significantly reduces MPS, with statistical significance indicated (**** *p* < 0.0001).

## Data Availability

All data used in this study were obtained from public databases. Single-cell RNA-seq (scRNA-seq) data were sourced from GEO (Gene Expression Omnibus, https://www.ncbi.nlm.nih.gov/geo/, accessed on 15 July 2024), OMIX (https://ngdc.cncb.ac.cn/omix/, accessed on 9 August 2024), ArrayExpress (https://www.ebi.ac.uk/biostudies/arrayexpress/, accessed on 2 September 2024), Single Cell Portal (https://singlecell.broadinstitute.org/single_cell, accessed on 15 September 2024), and SRA (https://trace.ncbi.nlm.nih.gov/Traces/sra/, accessed on 18 October 2024). Accession numbers are as follows: BC: GSE180286 [122], GSE225600 [123], GSE176078 [124], GSE264205 [125]; BLCA: GSE222315 [126], GSE135337 [127]; ccRCC: GSE210038 [128], GSE224630 [129], SCP1288 [130], SRZ190804 [131]; CRC: GSE245552 [132], GSE200997 [133]; ESCC: OMIX005710 [134]; GC: GSE206785 [135], GSE183904 [136], GSE246662 [137]; HCC: GSE149614 [138], GSE202642 [139]; HNSCC: GSE188737 [140], GSE185965 [141], GSE181919 [142], GSE206332 [143], GSE172577 [144]; LC: GSE171145 [145], E-MTAB-13526 [146]; OC: GSE184880 [147], GSE173682 [148], E-MTAB-8107 [149]; PDAC: GSE263733 [150], GSE154778 [151], GSE197177 [152]; PRAD: GSE181294 [153], GSE141445 [154]. Spatial transcriptomics data were all obtained from GEO, including: BC: GSE210616 [155] (GSM6433585); ccRCC: GSE175540 [156] (GSM5924031); PDAC: GSE203612 [157] (GSM6177618). Bulk RNA-seq data used for model construction and validation were obtained from TCGA and GEO, including: TCGA-BRCA, TCGA-BLCA, TCGA-KIRC, TCGA-COAD, TCGA-READ, TCGA-ESCA, TCGA-STAD, TCGA-LIHC, GSE148355 [158], TCGA-HNSC, TCGA-LUSC, TCGA-LUAD, TCGA-OV, TCGA-PAAD and TCGA-PRAD. For independent external validation, additional datasets were also collected from TCGA and GEO, including GSE3494 [159], GSE188715 [160], GSE167573 [161], GSE17536 [162], GSE149609 [163], GSE15459 [164], GSE76427 [165], GSE65858 [166], GSE30219 [167], GSE14407 [168], GSE79668 [169], GSE46691 [170], TCGA-CESC, and TCGA-UCEC. Proteomic datasets used for validation were retrieved from the PDC database, including PDC000120 [171], PDC000127 [172], PDC000116 [173], PDC000221 [174], PDC000234 [175], PDC000114 [176], PDC000270 [177], and PDC000504 [178].

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
