# Peer review of "Construction of Metastasis Prediction Models and Screening of Anti-Metastatic Drugs Based on Pan-Cancer Single-Cell EMT Features"

_ijms, 2025, doi:10.3390/ijms262311582_

Round 1

Reviewer 1 Report

Comments and Suggestions for Authors

Invasion and metastasis of malignant cells from their primary site to distant organs remain the leading causes of cancer-related mortality. Metastasis arises from a complex interplay of molecular pathways and dynamic interactions within the tumor microenvironment during tumor evolution. In this manuscript, titled “Construction of Metastasis Prediction Models and Screening of Anti-metastatic Drugs Based on Pan-Cancer Single-Cell EMT Features,” Xu et. Al., undertake a comprehensive pan-cancer analysis across 12 epithelial cancer types, covering 265 samples and over 1.2 million single cells, to derive both global and cancer-specific metastatic signatures. Using multiple machine-learning approaches, they generate metastasis prediction scores across cancers.

The study employs extensive dataset including several spatial transcriptomics datasets—and leverages a broad suite of computational tools to evaluate known biological features associated with epithelial–mesenchymal transition (EMT) and metastasis. The authors integrate datasets, annotate major cell populations, and examine EMT-related features in epithelial cells as well as non-epithelial populations such as cancer-associated fibroblasts (CAFs). Additional noteworthy aspects of the study include analysis of tumor-CAF interactions, characterization of the tumor immune microenvironment across metastatic stages, and exploration of therapeutic associations through candidate drug impact assessments.

Overall, the study is comprehensive, wide-ranging, and generally well written. However, few methodological clarity, conceptual ambiguities, and figure clarity limitations should be addressed.

Below I outline my major comments and suggestions.

Major Comments & Suggestions

  1. Unexpectedly high T/NK-cell fractions (Fig. 1D)

The reported T/NK-cell fractions appear unusually high for tumor-derived single-cell datasets, where immune cells typically constitute a smaller proportion due to enzymatic dissociation biases and epithelial enrichment. Please clarify whether this arises from differences in dissociation protocols, sample selection, isolation or data integration effects.

  1. Rationale and methodology for EMT scoring in non-epithelial cells

The manuscript discusses EMT score assessment across multiple non-epithelial cell types (fibroblasts, endothelial cells, immune cells), but the rationale and methodological sequence remain unclear.

How were the initial EMT scores computed, and how do they differ from the later high/mid/low (h/m/l) EMT classifications? The usage of WGCNA method is stated only later and the initial section states using scRNAseq data and TCGA RNAseq data for generating the EMT scores; please clarify whether EMT module genes used early in the analysis were derived independently of WGCNA. EMT gene programs are not specific to malignant epithelial cells, and elevated EMT signatures in fibroblasts or endothelial cells are not unexpected and do not necessarily indicate EMT relevant to metastasis.

Including immune cells in EMT scoring is conceptually problematic; EMT signatures in immune cells likely reflect transcriptional noise or context-specific activation rather than true EMT.

  1. EMT score differences between metastatic vs. non-metastatic groups

Would EMT signatures change significantly when stratifying malignant vs. non-malignant or metastatic vs. non-metastatic samples? This would reinforce the biological relevance of the EMT modules and help avoid confounding by stromal signals.

  1. Figure quality and readability

Many figures including histograms, UMAPs, and spatial plots are difficult to interpret due to small font sizes, low resolution, or poor color contrast. Reformatting and recoloring (or increasing resolution) is strongly recommended for clarity.

  1. CNV score definition and threshold rationale (Fig. S1DE)

CNV inference was performed to distinguish malignant from non-malignant epithelial cells, using thresholds of CNV score > 0.002 and correlation coefficient > 0.1. However: The CNV values displayed in Fig. S1D do not appear to match these thresholds. The figure legend states they represent “CNV scores,” but their scale is inconsistent with the stated cutoff (Box-Violin plots in the FigS1D). Please clarify what the plotted values represent and provide justification for choosing the thresholds.

  1. Visualization of malignant vs. non-malignant proportions

UMAPs showing malignant vs. non-malignant cell proportions is unclear. Stacked bar plots or ridge plots would communicate these proportions more effectively. Alternatively, recolor UMAPs for better visibility.

  1. EMT score trends across clinical stages (Fig. 2)

The manuscript states that EMT scores and the prevalence of hEMT cells increases with disease stage. However, in some cancer types EMT scores appear to decrease from stage III to IV. Please clarify whether this represents biological heterogeneity, sampling biases, or batch effects.

  1. CSC score correlations with EMT (Fig. S3DE)

Similarly, the correlation between the CSC scores and the EMT score (FigS3D and E) based classification also do not align with the statements made. The correlation between cancer stemness (CSC) scores and EMT-based classification does not clearly support the conclusions made.

  1. CytoTRACE interpretation in a pan-cancer context

CytoTRACE infers differentiation state using gene-count–based transcriptional promiscuity. In pan-cancer analyses, this can be confounded by: tissue-specific transcriptional baselines, lineage-intrinsic differences, cancer type-specific chromatin accessibility. Thus, comparisons across unrelated tumor types must be interpreted with caution. Additionally, EMT-associated plasticity is meaningful in epithelial tumors but may not apply to leukemias, lymphomas, or other non-epithelial cancers. Consider discussing these limitations.

  1. Use of both Monocle2 and Slingshot for pseudotime

The authors have used Monocle v2 and Slingshot for pseudotime trajectory analysis. Was there a particular reason for this, please clarify? Especially since they combined Slingshot pseudotime trajectory analysis followed by Monocle to perform differential expression analysis on hEMT cells and myCAFs.

  1. Spatial transcriptomics analysis requires significant figure improvement

Spatial figures lack sufficient resolution, color contrast, and labeling to support the conclusions drawn about spatial interactions between EMT subtypes and CAF subpopulations. Improvements to figure resolution and supplementary legends are needed for appropriate interpretation.

Overall Recommendation

The manuscript presents an ambitious and valuable integrative pan-cancer analysis of EMT features at single-cell resolution, with potential translational implications. However, several methodological ambiguities, conceptual concerns regarding EMT scoring in non-epithelial compartments, and figure clarity issues require significant clarification and revision. With appropriate methodological clarifications and figure improvements, this manuscript would be substantially strengthened.

Reviewer 2 Report

Comments and Suggestions for Authors

Summary: The manuscript titled “Construction of Metastasis Prediction Models and Screening of Anti-metastatic Drugs Based on Pan-Cancer Single-Cell EMT Features” by Xu et al interrogates features that predict metastasis and efficacy of anti-metastatic drug via construction of a computational predictive model in the context of epithelial to mesenchymal transition in across multiple cancer types. Xu et al developed a pan-cancer atlas via integration of 32 publicly available single-cell transcriptomic data sets, enabling identification of 8 major cell types including epithelial cells, B cells, plasma cells, myeloid cells, mast cells, fibroblasts and endothelial cells. Malignant cells were delineated via variability in copy number variation (Figure1). Xu et al then characterized transcriptomic features associated with metastasis and increased epithelial to mesenchymal transition (EMT) corresponded to increased tumor progression which was associated with increased expression of cancer stem cell features, TGFB signaling and glycolysis (Figure 2). 

Additionally, Xu et al characterized tumor associated fibroblast subsets and observed that increased levels of myofibroblastic cancer-associated fibroblasts predicted tumor progression (Figure 3). Furthermore, Xu et all developed a metastasis prediction model to integrate their findings into a translational framework; they observed that the intratumoral immune landscape predicted metastasis (Figure 4). To support identification of antimetastatic treatment approaches, Xu et al utilized a computational drug screening framework and delineated that across cancer types, Fostamatinib, a SYK inhibitor, exhibited the greatest antimetastatic potential (Figure 5); other compounds exhibited more restricted antimetastatic potential in specific tumor types.

In summary, the manuscript is interesting and well written. However, wet lab validation of these findings would significantly enhance the overall quality of this manuscript. 

  1. Specifically, transcriptional changes do not always result in changes in protein expression, underlining a need to integrate proteomic datasets into their approaches to fortify their conclusions.  
  2. In vitro and in vivo validation of the anti-metastatic potential of top compounds such as Fostamatinib would significantly enhance the impact of the reported computational approach.

Reviewer 3 Report

Comments and Suggestions for Authors

Dear Author,

Congratulations on your manuscript titled “Construction of Metastasis Prediction Models and Screening of Anti-metastatic Drugs Based on Pan-Cancer Single-Cell EMT Features.” This is a well-organized and insightful study in which you analyzed 12 cancer types and screened 1,217,505 cells using bulk RNA sequencing.

Major Comments:
The manuscript provides a comprehensive analysis of multiple gene sets; however, it does not consider autophagy-related genes (cell organelle specific, e.g., mitophagy) or key epigenetic regulators (including HDACs, HATs, and DNMTs). Given the critical roles these pathways play in cancer progression and metastasis, integrating them into your analysis would substantially strengthen the study and offer a more holistic view of the metastatic process.

Minor Comments:

  • The pathway labels in Figure 2D should be standardized for uniformity.

  • Figure legends should be expanded to clearly highlight the major findings.

  • The Methods section would benefit from more detailed descriptions of the analytical steps, with citations to relevant reference methods.

Overall, this is an impressive and thoughtfully executed study. Addressing the points above will further enhance the clarity, depth, and impact of your work.

Round 2

Reviewer 1 Report

Comments and Suggestions for Authors

Thanks for addressing my concerns and for considering my suggestions. I appreciate the appropriate changes and clarifications provided, which has sufficiently addressed my concerns.